

# 3D Radiative Heating of Tropical Upper Tropospheric Cloud Systems derived from Synergistic A-Train Observations and Machine Learning

Claudia J. Stubenrauch[1], Giacomo Caria[1], Sofia E. Protopapadaki[2], Friederike Hemmer[1]

[1]Laboratoire de Météorologie Dynamique / Institut Pierre-Simon Laplace, (LMD/IPSL), Sorbonne Université, Ecole Polytechnique, CNRS, Paris, France
[2]COOPETIC, Paris, France

*Correspondence to*: Claudia J. Stubenrauch (stubenrauch@lmd.polytechnique.fr)

**Abstract.** Upper Tropospheric (UT) cloud systems constructed from Atmospheric Infrared Sounder (AIRS) cloud data provide a horizontal emissivity structure, allowing to link convective core to anvil properties. By using machine learning techniques we composed a horizontally complete picture of the radiative heating rates deduced from CALIPSO lidar and CloudSat radar measurements, which are only available along narrow nadir tracks. To train the artificial neural networks, we combined the simultaneous AIRS, CALIPSO and CloudSat data with ERA-Interim meteorological reanalysis data in the tropics over a period

of four years. Resulting non-linear regression models estimate the radiative heating rates as a function of about 40 cloud, atmospheric and surface properties, with a column-integrated mean absolute error (MAE) of 0.8 K/d (0.5 K/day) for cloudy scenes and 0.4 (0.3 K/day) for clear sky in the longwave (shortwave) spectral domain. Already about 20 basic input variables yield good results, with a 6% (10%) larger MAE. Developing separate models for i) high opaque clouds, ii) cirrus, iii) mid- and low-level clouds and iv) clear sky, independently over ocean and over land, lead to a small improvement, when considering

the profile shapes. These models were then applied to the whole AIRS cloud dataset, combined with ERA-Interim, to build 3D radiative heating rate fields. Over the deep tropics, UT clouds have a net radiative heating effect of about 0.3 K/day throughout the troposphere from 250 hPa downward, with a broad maximum of about 0.4 K/d around 330 hPa, enhancing the column-integrated latent heating by about 25%. This value is larger than earlier results of about 20%. Above the height of 200 hPa, the LW cooling above convective cores and thick cirrus anvils is opposed by thin cirrus heating. Whereas in cooler regions

low-level clouds also influence the net radiative heating profile, in warmer regions it is nearly completely driven by deep convective cloud systems. These mesoscale convective systems (MCS) are colder and include slightly more thin cirrus around their anvils than those in cooler regions. Hence, the MCSs over these warmer regions produce a vertically more extended heating by the thicker cirrus anvils and a heating of 0.7 K/d above the height of 200 hPa by the surrounding thin cirrus. The roughly estimated horizontal gradients between cirrus anvil and convective core as well as between surrounding thin cirrus and cirrus anvil seem to be slightly smaller in warmer regions, which can be explained by their larger coverage. The 15-year

time series of the heating / cooling effects of MCSs are well related to the ENSO variation. While the coverage of all MCSs is



relatively stable (or very slightly decreasing) with surface warming, with -1.3 ± 0.6 %/K, the coverage of cold MCSs relative to all MCSs significantly increases by +18 ± 5 %/K.

## 1 Introduction

Upper tropospheric (UT) clouds play a vital role in the climate system by modulating the Earth's energy budget and the UT heat transport. These clouds cover about 30% of the Earth and even 40% of the tropics (e. g. Stubenrauch et al. 2013, 2017). Yet, their role in the climate change feedback is still highly uncertain (e. g. Boucher et al., 2013, Zelinka et al., 2016). Tropical organized deep convection leads to cloud systems with stratiform cirrus anvils of the size of several 1000's km$^2$ (e.g. Houze, 2004). Living much longer than the convective towers themselves, these cirrus anvils produce a radiative heating that is

expected to be as important for the large-scale circulation as the released latent heat in the initial stage of convection. In tropical convective regions more than 50% of the total heating is contributed by cirrus radiative heating (e.g. Sohn 1999). This heating, induced by the anvils and cirrus, then influences the large-scale tropical atmospheric circulation (e.g. Slingo and Slingo, 1991; Sherwood et al., 1994). It is affected by: i) the areal coverage, ii) the horizontal cloud emissivity structure within the systems, and iii) the vertical structure of the cirrus anvils (layering and microphysics). The influence of the vertical distribution of

radiative heating was demonstrated  on large-scale tropical circulations by Stephens and Webster (1984) and Bergman and Hendon (2000) and on the local cloud structure by Mather et al. (2007). The net radiative heating associated with tropical anvils and cirrus layers is also known to play a major role in the thermodynamic stability of the upper troposphere (Ackerman et al., 1988) and self-regulation of tropical convection (e.g. Stephens et al., 2004, 2008).

So far, observational studies of tropical mesoscale convective systems (MCSs) have concentrated on the convective towers

and the thick cirrus anvils (e.g. Yuan and Houze 2010, Roca et al. 2014). Yet thin cirrus correspond to about 30% of / around the anvil area of the deep convective systems (Protopapadaki et al. 2017). Other studies, focusing on their vertical structure along narrow nadir tracks (Fig. 1), missed the lateral horizontal dimension (e.g. Igel et al., 2014; Stein et al. 2017). The organisation of convection was studied by statistical analysis of "cloud regimes", defined by similar cloud property distributions within grid cells (e.g. Tselioudis et al. 2013, Tan et al. 2015, Oreopoulos et al., 2016). Suggesting a connection between

radiative effects and dynamics, this concept is very valuable, but it misses the horizontal extent of the systems.

A study by Li et al. (2013) finds that the column-integrated radiative heating of tropical UT clouds accounts for about 20% of the latent heating. The radiative heating was estimated by combining International Satellite Cloud Climatology Project (ISCCP) data, classified as four distinct cloud regimes at a spatial resolution of 2.5° latitude and longitude, with heating rate profiles assigned from two tropical Atmospheric Radiation Measurement (ARM) sites, while the latent heating was deduced

from measurements of the Tropical Rainfall Measuring Mission (TRMM) Precipitation Radar (PR). However, ISCCP and ARM data both may underestimate the effect of thin cirrus, because its occurrence may be missed by ground observation





(Protat et al., 2014) and by ISCCP (e.g. Stubenrauch et al., 2013), in particular when low-level clouds are also present and during night.

Therefore, to include also the thinner cirrus and the complete 3D structure of these cloud systems, we applied a different

strategy: To estimate the radiative heating rates of UT clouds we combined observations which are more sensitive to thin cirrus, together with machine learning techniques and a cloud system approach. The good spectral resolution of IR sounders makes them sensitive to cirrus, down to a visible optical depth of 0.2, during daytime and nighttime. Cloud properties retrieved from measurements of the cross-track scanning Atmospheric Infrared Sounder (AIRS) aboard the polar orbiting Aqua satellite have a large instantaneous horizontal coverage (Stubenrauch et al., 2017). They have been used by Protopapadaki et al. (2017)

to reconstruct UT cloud systems. Recently these datasets have been extended, so that they now cover Sep 2002 to Aug 2019. On the other hand, the space-borne active lidar and radar measurements of the CALIPSO and CloudSat missions (Stephens et al., 2002) supply the cloud vertical structure, in particular the radiative heating rates (Henderson et al., 2013). As this information is only available along successive narrow nadir tracks, separated by about 2500 km, we employed machine learning techniques on cloud, atmospheric and surface properties to build a 3D description of these cloud systems. These

techniques were already successfully applied to extend IR brightness temperature (Kleynhans et al., 2017) and snow water (Snauffer et al., 2018) from other atmospheric variables.

This article presents the effect of UT clouds on tropical radiative heating rates in the longwave (LW) and shortwave (SW) spectral domain and the relationship between surface temperature, convective depth and anvil radiative heating / cooling. Data and Methods are described in Section 2. Sensitivity studies and evaluation of the developed non-linear regression models via

machine learning are presented in Section 3. They give insight into the most appropriate cloud and atmospheric properties as well as on how many scene-dependent non-linear regression models are necessary to reliably predict the radiative heating rates of different cloud types and of clear sky, over ocean and land. After application of these models to the AIRS cloud and ERA-Interim atmospheric and surface data, section 4 investigates the contribution of clouds and in particular of thin cirrus and MCSs on the radiative heating / cooling. Conclusions and outlook are given in Section 5.

**2 Data and Methods**

**2.1 AIRS Cloud Data and Cloud System Data**

Since 2002 AIRS (Chahine et al., 2006) aboard the National Aeronautics and Space Administration (NASA) Earth Observation Satellite Aqua provides very high spectral resolution measurements of Earth emitted radiation in the thermal IR (3.74 ó 15.40 μm) at 1:30AM and 1:30PM local time (LT). Cross-track scanning leads to a large instantaneous coverage of about 70% in

the tropics. The spatial resolution of these measurements at nadir is about 13.5 km. Nine AIRS measurements (3 x 3) correspond to one footprint of the Advanced Microwave Sounder Unit (AMSU).



The Clouds from IR Sounders (CIRS) data (Stubenrauch et al., 2017) provide cloud pressure ($p_{cld}$), cloud emissivity ($\varepsilon_{cld}$), as well as cloud temperature ($T_{cld}$) and cloud height ($z_{cld}$), together with their uncertainties. The cloud retrieval is based on a weighted $\chi^2$ method (Stubenrauch et al., 1999), which uses eight channels along the 15 μm $CO_2$ absorption band, with peak

contributions coming from 235, 285, 375, 415, 565, 755, 855 hPa and near the surface. Minimizing $\chi^2$ leads to a pair of $p_{cld}$ and $\varepsilon_{cld}$ values. The uncertainty on $p_{cld}$ is estimated by identifying the pressure level at which the normalized weighted $\chi_{norm}^2(p)$ = $(\chi^2(p) - \chi_{min}^2)/(\chi_{max}^2 - \chi_{min}^2)$ reaches 0.025, a value empirically determined by Feofilov et al. (2017). The uncertainty on $\varepsilon_{cld}$ is then determined as the difference between $\varepsilon_{cld}$ and the value of $\varepsilon_{cld}$ computed at this pressure level. UT (high-level) clouds are defined as clouds with $p_{cld} < 440$ hPa. UT cloud types are further distinguished with respect to $\varepsilon_{cld}$ as opaque high clouds

(Cb, $\varepsilon_{cld} \geq 0.98$), cirrus (Ci, $0.98 < \varepsilon_{cld} \leq 0.5$) and thin cirrus (thCi, $0.5 < \varepsilon_{cld} \leq 0.1$). $p_{cld}$ is transformed to $T_{cld}$ and $z_{cld}$ via the atmospheric temperature and water vapour profiles of the ancillary data (see section 2.2). An 'a posteriori' multi-spectral cloud detection is based on the spectral coherence of retrieved cloud emissivity in the atmospheric window between 9 and 12 μm. This spectral region also provides information on the thermodynamic phase of the clouds, and for semi-transparent cirrus the slope of cloud emissivities between 9 and 12 μm gives an indication of the effective ice crystal diameter (Guignard et al. 2012).

To get information on the surrounding cloud scene structure within a region of 2° latitude x 2° longitude, we distinguished sixteen cloud regimes by applying a k-means clustering on histograms of $\varepsilon_{cld}$ and $p_{cld}$ from CIRS-AIRS data, similar to the method developed by Rossow et al. (2005) using ISCCP data. In addition, we provide the clear sky fraction estimated from AIRS within these grid cells.

The reconstruction of UT cloud systems is based on two independent variables, $p_{cld}$ and $\varepsilon_{cld}$ (Protopapadaki et al. 2017): The

AIRS cloud data were first merged to grid cells of 0.5° latitude x 0.5° longitude, and only grid cells with more than 70% UT clouds were used to reconstruct UT cloud systems. After having filled the data gaps between adjacent orbits, UT cloud systems were built from adjacent elements of similar cloud height, given by $p_{cld}$; in a second step convective cores, thick cirrus and thin cirrus were identified within the anvils by using $\varepsilon_{cld}$. This cloud system approach was used in section 4 to identify MCSs and to relate the radiative heating / cooling profiles of their convective cores and their anvils to different surface conditions.

Therefore MSCs were defined as UT cloud systems with at least one convective core (built from grid cells with average $\varepsilon_{cld} > 0.98$ within subregions of $\varepsilon_{cld} > 0.9$) and a core fraction within the system of at least 0.05.

## 2.2 Atmospheric and surface data

ERA-Interim (Dee *et al.* 2011) is a global atmospheric reanalysis provided by the European Centre for Medium-Range Weather Forecast (ECMWF). Land surface, oceanographic, atmospheric, and spaceborne measurements from numerous sources are

assimilated in the Integrated Forecast System. The data assimilation scheme is sequential: at each time step, it exploits available observations to constrain the model, which in turn provides a short-range forecast for the next assimilation time step. Gridded



data products (at a spatial resolution of 0.75° latitude x 0.75° longitude) include 6-hourly surface temperature, atmospheric temperature and water vapour profiles, as well as dynamical parameters such as horizontal and vertical large-scale winds. These data are given at Coordinated Universal Time (UTC) values of 0:00, 6:00, 12:00 and 18:00. In order to match these

data with the AIRS observations, the former were interpolated to the corresponding local time, using a cubic spline function, as in Aires *et al.* (2004). Surface temperature and pressure, atmospheric profiles of temperature and water vapour, interpolated to 23 pressure levels, and tropopause pressure determined from the temperature profile according to Reichler et al. (2003) were added, since the CIRS retrieval relies on these ancillary data for the retrieval (Stubenrauch et al., 2017). We then derived the relative humidity within the 22 atmospheric layers from the temperature and water vapour profiles by a method based on

(Stubenrauch and Schumann, 2005). The CIRS cloud retrieval classifies the temperature and water vapour profiles by comparing them to about 2300 representative clear sky atmospheric profiles of the Thermodynamic Initial Guess Retrieval (TIGR) data base (Chédin et al. 1985, 2003), which includes the spectral atmospheric transmissivities used in the radiative transfer part of the retrieval. This atmospheric classification provides additional information for the non-linear regression models developed in section 3.

Spectral IR surface emissivities over land have been retrieved in the case of clear sky conditions using measurements of the IR Atmospheric Sounding Interferometers (IASI) by Paul et al. (2012). We use a monthly mean climatology at a spatial resolution of 0.25° x 0.25° for wavelengths with values around 9.00, 10.16and 12.18 μm. The surface emissivity of water is set to 0.99 for 9 μm and to 0.98 for the other wavelengths, according to Wu and Smith (1997).

For the prediction of the SW heating rates during daytime we use, instead of the spectral IR surface emissivities, the visible

surface albedo at noon local solar time and the solar zenith angle. The land surface albedos, retrieved under clear sky conditions at wavelengths of visible light measured by the Moderate Resolution Imaging Spectroradiometer (MODIS) sensors on NASAøs Aqua and Terra satellites (MODIS Collection 5, MOD43 product, Strahler et al., 1999), are distributed as a monthly climatology at a spatial resolution of 0.1° x 0.1° by the NASA Earth Observations (NEO) website (https://neo.sci.gsfc. nasa.gov/). Over ocean we assume a surface albedo at noon local solar time of 0.06.

MODIS also provides aerosol optical depth (AOD) at a wavelength of 550 nm (MODIS Collection 5, MOD04/MYD04 product, Levy et al., 2009). We explore the benefit of adding the AOD of a monthly climatology at a spatial resolution of 0.25° x 0.25°, also provided by the NEO website, to the input variables to be used by the machine learning.

Finally we explore the value of adding the vertical velocity at 500 hPa, at the spatial resolution of 0.375° x 0.375°, from the ERA5 reanalysis (Hersbach et al., 2020).

**2.3 CALIPSO-CloudSat vertical structure and collocation with AIRS**

The vertical structure of the clouds can only be determined by active spaceborne instruments. The Cloud-Aerosol Lidar with Orthogonal Polarization (CALIOP) aboard CALIPSO and the Cloud Profiling Radar (CPR) aboard CloudSat, both part of the



A-Train constellation, follow AIRS within a few minutes. CALIOP (Winker et al., 2009) provides backscatter profiles at a wavelength of 532 nm and 1064 nm, with a vertical and horizontal resolution of about 30 m and 90 m, respectively.

Measurements are sampled every 333 m along the track. The backscatter ratio helps to distinguish between aerosols and clouds. The 94 GHz nadir-viewing CPR (Stephens et al., 2002) measures profiles of the power backscattered by clouds at a native vertical resolution of 480 m over footprints covering 1.8 km×1.4 km. By using oversampling, data are provided at a vertical resolution of 240 m. While CALIOP is even sensitive to subvisible cirrus, its signal only penetrates clouds to a level where the cloud gets opaque (i. e. the optical depth reaches about 3). On the other hand, CPR is able to probe optically thick cloud

layers and therefore provides their correct cloud base. By combining information from both instruments it is possible to describe completely the vertical structure of clouds. However, this information is only given along successive nadir tracks. The radiative fluxes and heating rates of 2B-FLXHR-LIDAR (version R04; Henderson et al., 2013; L'Ecuyer et al., 2008) were derived by applying the BUGSrad broadband radiative transfer model (Ritter and Geleyn, 1992) to the scenes observed by CALIPSO-CloudSat, using as inputs the vertical location of the cloud layers (2B-GEOPROF-LIDAR), the cloud water /

ice content and effective particle sizes retrieved from radar only (2B-CWC-RO) (Austin et al., 2009), distinction between cloud and rain water contents from 2C-PRECIP-COLUMN (Haynes et al, 2009) and collocated atmospheric and surface auxiliary data from ECMWF. For the clouds and aerosols which are undetected by CloudSat, the MODIS-based cloud optical depth (2B-TAU) and CALIPSO version-3 products (Trepte et al., 2010) are used to calculate the corresponding radiative properties. Thin clouds which are only detected by CALIPSO are defined as ice clouds by a temperature smaller than 253.15

K, and their ice crystal equivalent mass sphere effective radius is assumed to be 30 μm for the radiative transfer calculations. According to a study of Protat et al. (2014) over the tropical ARM site of Darwin, the shapes of the radiative heating rates are in good agreement with those derived from ground-based remote sensing by radiative transfer calculations (McFarlane et al., 2013) and with those from an experimental 2C-ICE-FLUX product, between altitudes of 1 and 12 km. Above 12 km, ice properties included from 2C-ICE lead to a reduced LW cooling and enhanced LW warming of about 0.3 K/day compared to

2B-FLXHR-LIDAR, whereas the underreported cirrus frequency by the ground-based lidar at the ARM site leads to an enhanced LW cooling and reduced LW warming of about 0.4 to 0.8 K/day. The 2C-ICE-FLUX product assumes different cloud microphysical and radiative properties than the 2B-FLXHR-LIDAR data, in particular the ice microphysical properties are obtained from 2C-ICE (Deng et al., 2013), when 2B-CWC-RO has no data. The same microphysical properties, together with improved cloud phase identification and surface characteristics, are

integrated in the very recently released version R05 of FLXHR-LIDAR data (Matus and L'Ecuyer, 2017). The improvements led to a better agreement with TOA fluxes from the Clouds and the Earth's Radiant Energy System (CERES), with global annual mean TOA net cloud radiative effect of -17.1 ± 4.2 Wm$^{-2}$ (Matus and L'Ecuyer, 2017) compared to -18.1 ± 3.7 Wm$^{-2}$ (Henderson et al., 2013) and global annual mean atmospheric cloud radiative effect between both versions differing by about 10% (Hang et al., 2019): 7.8 Wm$^{-2}$ (R05) compared to 8.6 Wm$^{-2}$ (R04).





The comparison of 2B-FLXHR-LIDAR (R04) with CERES-CALIPSO-CloudSat-MODIS (CCCM) products, using a finer vertical resolution and different microphysics than FLXHR-LIDAR, revealed a small low bias in SW heating of FLXHR-LIDAR due to a slight underestimation of cloud occurrence of height below 1 km, while the LW heating of CCCM for thin cirrus may be slightly high biased (Ham et al., 2017).

We extended the collocated AIRS-CALIPSO-CloudSat data used by Feofilov et al. (2015) and Stubenrauch et al. (2017) by

the NASA 2B FLXHR-LIDAR (R04) heating rates for the period of 2007 to 2010. These are given at a vertical resolution of 240 m, corresponding to about 80 values over a height of 20 km. Since the AIRS cloud height is retrieved as pressure and the input parameters are probably not precise enough to predict such a fine vertical structure, we transformed the FLXHR-LIDAR heating rates to 22 pressure layers between 70 hPa and the surface. For each of the AIRS footprints this collocated dataset also includes the number of detected cloud layers, as well as cloud top and cloud base of each of these cloud layers, detected by

lidar or radar, at the spatial resolution of the radar footprints from the 2B-GEOPROF-LIDAR data. In section 3 we use the number of detected cloud layers to evaluate the clear sky identification by AIRS alone.

As version R05 of the FLXHR-LIDAR data was only released when we were finishing our analyses of section 4, we present the results which used 2B-FLXHR-LIDAR (R04) data for the training of the artificial neural networks (ANN), keeping in mind that the cirrus HRs above a height corresponding to 200 hPa are more reliable than those from ground-based measurements by

may be still slightly underestimated compared to newer versions with different ice microphysics (Protat et al., 2014; Ham et al., 2017; Hang et al., 2019). Within the overall uncertainties described in this section and in section 3, the results in Section 4 are still valid. In the future we will train the ANN models again with the improved version of the FLXHR-LIDAR data and a new version of the CIRS data (using ERA5 ancillary data, as ERA-Interim data production ceased in August 2019).

**2.4 Artificial Neural Network Construction**

The challenge in creating a complete 3D description of the UT cloud systems and their environment lies in the lateral expansion of the information on the vertical structure, only available at the locations sampled along the lidar-radar nadir tracks, illustrated in Figure 1. In order to achieve this goal, we developed nonlinear regression models based on ANNs on the combined AIRS and ERA-Interim data as described in sections 2.1 and 2.2. ANNs have seen spectacular progress during the last few years, especially in the automation of finding the most appropriate weights used in the ANN layers. We used the TensorFlow

framework (https://www.tensorflow.org) to train machine learning models available in the Keras program library (https://keras.io) for Python, with training and testing along the nadir tracks, using four years of collocated data.

Kleynhans et al. (2017) demonstrated that thermal IR radiation at top of atmosphere, as measured by MODIS, can be best simulated from available atmospheric reanalysis data by using a multi-layer perceptron (MLP) supervised learning technique. This technique produced the lowest overall error rates, in particular over cloudy situations, compared to non-linear support

vector regression (SVR), convolutional neural network (CNN) and even to atmospheric radiative transfer simulations. After





having explored the performance on the number of hidden layers within the MLP ANN, our final ANN consists of an input layer with the approximately 30 to 45 input variables (see section 3), two hidden layers with 64 neurons and an output layer which corresponds to the radiative heating rates given in 22 pressure layers. Hsieh (2009) states that often a single hidden layer is already adequate to model any nonlinear continuous function. To improve the performance, we used the rectified linear unit

(ReLU) activation function, corresponding to max(0,x). For a better efficiency we use the Adaptive Moment Estimation (Adam) optimizer, using adaptive learning rates (Kingma and Ba, 2014).

The training dataset is randomly separated into three portions: 80% are used for training, 10% for validation and 10% for testing. In order to have similar cloud type, day-night and ocean-land statistics in these portions, we stratified the data by cloud type, ocean-land and day-night for LW and by cloud type and ocean-land for SW (only available during daytime). We used

the mean absolute error (MAE) between the prediction and the target value from the lidar-radar observations as metrics for the quality of the regression model.

The input variables have first been normalized by subtracting the mean and dividing by the standard deviation over the four-year dataset. However, this metrics is difficult to use in phase 2, when we apply the model to all data, because in this case we would need to re-compute the mean and standard deviation of each variable over the whole data period, as they may change

with time. In addition, not all input variable distributions are Gaussian. Therefore, once we had established sensitivity tests to estimate the most appropriate variables, we trained final scene-dependent models by standardizing the input variables, which means subtraction of an acceptable minimum and division by the difference between acceptable maximum and minimum. These acceptable minimum and maximum values have been established for each variable and adapted to the scenes for which the models were trained: ocean or land, all cloud types, clear sky, high clouds or mid- and low-level clouds. Before the

application of the model, the input variables have been bounded between minimum and maximum in order to avoid outliers.

## 3 Evaluation and Sensitivity Studies

In this section we assess the sensitivity of the predicted radiative heating rates (HRs) to two different factors: the selection of input variables and the number of scene types for which the models are developed. The following cloud scenes are considered: i) all clouds over the whole tropical band, ii) all clouds separately over ocean and over land, iii) high-level clouds and

independently mid- / low-level clouds, each of both cloud types separately over ocean and over land and iv) Cb, Ci / thin Ci, mid- / low-level clouds independently, where each cloud type is further separated over ocean and over land. Clear sky is treated independently, because the cloud properties are not used in this case. For the application to all data, the cloud and clear sky models are then combined to construct the radiative HRs over the whole tropical band (section 3.3).





### 3.1 Sensitivity to input variables

The input variables describing the cloud, atmosphere and surface properties used for the prediction of the radiative HRs are summarized in Table 1.

The training for the SW HRs is based only on data recorded at 1:30PM LT, while the training for the LW HRs exploits data for both 1:30AM and 1:30PM. Since the CALIPSO data are slightly more sensitive during night-time, we used for the LW training a day-night flag as additional input variable. The choice of input variables slightly differs for the prediction of LW

and SW HRs: For the training of LW HRs, we used surface spectral IR emissivities while for the training of SW HRs we used surface albedo and solar zenith angle.

The MLP regression models compute from about 40 input variables radiative HRs for 22 pressure layers from 70 hPa to 1000 hPa. Earth topography implies that the temperature, relative humidity and radiative HR profiles are not always determined over all 22 pressure layers. Given that neural networks need a constant number of input and output values, we had to replace

the missing values below the surface. Therefore, we first continued the temperature, relative humidity and radiative HR profiles below $p_{surf}$ with their lowest valid value, and then added to these values the average vertical gradients between the corresponding layer and the layer with the lowest valid value. These gradients were computed using the average profiles of regions containing all 22 pressure layers, separately determined over ocean and over land, and per cloud type and month. Even if these values below the surface are not used in the analyses, they slightly influence the training.


**Table 1: List of input variables for the prediction of LW and SW heating rates.**

| ***Clouds*** | | |
|---|---|---|
| CIRS cloud properties and uncertainties | $\varepsilon_{cld}$, $p_{cld}$, $T_{cld}$, $d\varepsilon_{cld}$, $dp_{cld}$, $dT_{cld}$, $\chi_{min}^2$ | |
| cloud spectral emissivity difference | $(\varepsilon_{cld}(12\mu m) - (\varepsilon_{cld}(9\mu m))$ | |
| CIRS cloud regime (CR) at 2° x 2° | CR (1-16), kernel distance | |
| ***Atmosphere*** | | |
| AIRS $T_B$ at 0.5° x 0.5° | $T_B(11.85\mu m)$, $\sigma(T_B)$, $T_B(7.18\ \mu m)$ | |
| ERA-Interim atmospheric properties | TIGR atmosphere (1-1500), total precipitable water, $p_{tropopause}$ | |
| ERA-Interim relative humidity profile | RH (determined from T and water vapour) within 10 layers | |
| ERA-Interim temperature profile | T within 10 layers | |
| ERA5 vertical velocity | $\omega$ at 500hPa | |
| MODIS aerosol optical depth | AOD (monthly mean climatology) | |
| ***Surface*** | | |
| ERA-Interim surface properties | $p_{surf}$, $T_{surf}$, nb of atm. layers down to $p_{surf}$ | |
| IASI spectral surface emissivity | $\varepsilon_{surf}(9, 10, 12\mu m)$ | (monthly mean climatology) |
| surface albedo | $\alpha_{surf}$ | (monthly mean climatology) |
| solar zenith angle, day-night flag, land-ocean flag | | |





For the sensitivity study of the most appropriate variables, we considered cloudy scenes over ocean, and we set up six different

experiments to predict the LW (SW) HRs, starting with

1) a set of 18 (19) basic variables, which describe cloud, atmospheric and surface properties: CIRS cloud properties and their uncertainties, cloud spectral emissivity difference between 9 and 12 μm, AIRS brightness temperatures, total precipitable water, tropopause height and TIGR atmosphere class, surface pressure and temperature.

Then we gradually added to the basic variables of experiment 1:

2) weather state classification and its uncertainty given by the kernel distance: total of 20 (21) input variables,

3) relative humidity in ten layers: total of 30 (31) input variables,

4) atmospheric temperature in ten layers: total of 40 (41) input variables,

5) vertical velocity from ERA5 reanalyses: total of 41 (42) input variables, and

6) monthly mean aerosol optical depth: total of 42 (43) input variables.

Table 2 compares the mean absolute error (MAE) for the prediction of LW and SW heating rates of clouds over ocean from the experiments 1 to 6. In all cases the MAE over the validation data and over the testing data are within 0.01 K/d. Therefore only the MAE over the validation dataset is shown. We provide further details in Figure S1 of the supplement, presenting the MAE as a function of number of iterations (epochs). The similarity in MAE between the validation and testing data means that there is no under-fitting (the variables are not sufficient to predict the target) nor over-fitting (the model is too detailed, with

too many variables or the data base is not sufficiently large). A similar figure for the evaluation of the clear sky models is given in Figure S2 of the supplement. As shown in Table 2, the MAE decreases by about 5% (10%) for the LW (SW) model when the atmospheric profiles are included, while the addition of vertical velocity and AOD do not seem to improve the results. This lack of improvement may be explained by noise coming from these sources in combination with the AIRS cloud properties and ERA-Interim atmospheric and surface properties. The addition of the temperature profile only slightly improves the

prediction of the heating rates, most probably because the atmospheric T profiles are more similar within the tropics than the atmospheric relative humidity profiles.

**Table 2: MAE (K/d) for the prediction of LW or SW heating rates of clouds over ocean, from experiments 1 - 6.**

| ocean | basic | + WS | + RH10 | RH-T10 | + w500 | + AOD |
|-------|-------|------|--------|--------|--------|-------|
| **LW HR** | 0.84 | 0.84 | 0.80 | **0.79** | 0.79 | 0.79 |
| **SW HR** | 0.51 | 0.50 | 0.46 | **0.45** | 0.45 | 0.45 |





As the MAE only provides an average estimation of the quality of the prediction, we also considered the difference between the predicted radiative HRs and those determined from CALIPSO-CloudSat measurements over tropical ocean, separately for five cloud types: high opaque (Cb), cirrus (Ci), thin cirrus, midlevel and low-level clouds. The LW and SW results are compared between the different experiments in Figure S3 of the supplement. Overall, all results show good agreement between predicted and CALIPSO-CloudSat derived HRs. The differences between mean predicted and ±observedø radiative HRs

undulate well around 0 K/day. However, we note that when using the ERA5 vertical velocity at 500 hPa as an additional input variable, the results for Cb and mid- and low-level clouds in the LW are slightly degraded. Similarly, the addition of the monthly mean AOD does not improve the results. This indicates only a medium compatibility between these two variables and the instantaneous AIRS cloud properties and ERA-Interim atmospheric and surface properties. Therefore we use in the following the input variables of experiment 4 for the model development. The 30% quantiles and 70% quantiles of the HR

differences in Figure S3 give an indication of the uncertainty, which may be related to differences in horizontal resolution between AIRS and CALIPSO-CloudSat. In particular for convective towers of very large optical depth (Cb) and for mid- and low-level clouds, the coarse AIRS spatial resolution may lead to a mixture of several cloud types or of clouds and clear sky within one footprint.

### 3.2 Sensitivity to the selection of scenes used for the training

The second part of sensitivity studies is dedicated to the scene types for which we develop the models: i) all clouds over the whole tropical band (one model), ii) all clouds separately over ocean and over land (two models), iii) high-level clouds and mid- / low-level clouds individually, each separately over ocean and over land (four models), and iv) Cb, Ci / thin Ci, mid- / low-level clouds individually, each separately over ocean and over land (six models). In addition, we develop models for clear sky i) over the whole tropical band (one model) and ii) separately over ocean and over land (two models). In general, a model

trained over all scenes together will smooth out differences between different cloud types and between ocean and land. Also scenes which are less frequent may have a smaller weight and may be therefore less represented than other scenes. Since we are interested in the study of the effect of UT cloud systems, we choose to use separate models. In particular, the modelling of Cb clouds is improved when exploiting a dedicated training for this cloud type, which represents about 7% of all clouds in the deep tropics (Stubenrauch et al., 2017).

When using one model for all clouds over ocean and land, the MAE is 0.82 K/day for LW and 0.51 K/day for SW HRs. Table 3 presents the MAE for the prediction of LW and SW HRs over the validation data, separately for different scene types over ocean and over land. In general, the performance is slightly better over ocean than over land, which can be explained by a greater homogeneity of surface, in particular in the SW, and atmospheric properties. We also observe a decreasing performance




from clear sky scenes (LW 0.36 K/day and SW 0.27 K/day) over mid- / low-level clouds towards high-level clouds and Cb,
which again can be explained by an increasing inhomogeneity, and in the case of Cb the saturation of $\varepsilon_{cld}$ at 1.

To illustrate the effect of model aptness in dependence of training scenes, we compare in Figure 2 the difference between the
predicted radiative HRs and those from CALIPSO-CloudSat over ocean for Cb, cirrus and thin cirrus, in the LW and SW,
respectively. Compared are models which were trained i) for all clouds over ocean and land together, ii) for all clouds over
ocean, iii) for high-level clouds over ocean and iv) for Cb and for Ci / thin Ci over ocean. All results are quite similar, with the
differences between mean predicted and ÷observedø radiative HRs undulating well around 0 K/day. However, we observe an
overestimation of the LW cooling above Cb clouds by nearly 1 K/d when all clouds together are used to develop one single
model. The results improve for cirrus and thin cirrus when a dedicated model is developed for these cloud types. For the SW
HRs it is not possible to determine the best performance among these four models. The SW heating in the upper part of Cb
clouds is more difficult to predict, as for all four models the mean difference undulates around 0 K/day within ± 0.8 K/day
between 100 and 200 hPa. Considering the radiative HR profiles shown in Figure S4 of the supplement, we find that the largest
uncertainties for Cb clouds are around the maxima of LW cooling and SW heating. Furthermore, we observe that all models
give very similar results, so that in the following we will mainly use the most specific scene models, leading to the application
of eight models to reconstruct the radiative heating rate fields over the tropics.

**Table 3: MAE (K/d) for the prediction of LW and SW heating rates over different scene types.**

| ocean | clouds | high | Cb | cirrus | mid/low | clear |
|-------|--------|------|------|--------|---------|-------|
| **LW HR** | 0.79 | 0.91 | 1.10 | 0.90 | 0.69 | 0.34 |
| **SW HR** | 0.45 | 0.62 | 1.10 | 0.59 | 0.33 | 0.22 |
| **land** | clouds | high | Cb | cirrus | mid/low | clear |
| **LW HR** | 0.88 | 0.99 | 1.24 | 0.97 | 0.67 | 0.39 |
| **SW HR** | 0.69 | 0.77 | 1.35 | 0.72 | 0.54 | 0.36 |

### 3.3 Construction of Tropical Heating Rate Fields

We then applied the ANN models to one month of AIRS data, January 2008, corresponding to a La Niña condition. During
this condition, tropical convection is shifted towards the West Pacific. In order to obtain the radiative HRs over the whole
tropical band, the different models were applied to the corresponding scenes. The results were then averaged at a spatial



resolution of 0.5° latitude x 0.5° longitude. This allows us to compare the results of the laterally extended radiative HRs with those of FLXHR, providing statistics only along the nadir tracks.

Figure 3 shows the contrast between the mean radiative LW and SW HRs of five cloud types (Cb, Ci, thin Ci, mid- and low-level clouds) and the one related to clear sky, as identified from AIRS, over the latitude band 30N ó 30S. For comparison, the

radiative HRs from FLXHR, averaged over the nadir tracks, (broken lines) are also shown. Averages of predicted and ÷observedø radiative HRs are very similar, despite different sampling and spatial resolution. This means that the nadir track statistics gives a good picture on monthly average over the whole tropics and that the prediction models provide reliable results. In a clear sky situation, LW cooling occurs, linked to the absorbed and transmitted energy by the molecules in the atmosphere. As shown in Figure 3, this cooling decreases from about -2.5 K/day with increasing height until it reaches about 0 K/day

around 100 hPa. Its variability over the tropics is small. Since the AIRS clear sky identification may also include subvisible cirrus as well as partly cloudy scenes within the AIRS footprint, we estimated how much this affects the radiative HRs by comparing the FLXHR HRs for AIRS clear sky and for CALIPSO-CloudSat clear sky identification. As shown in Figure S5 of the supplement, one observes a slight positive bias in the clear sky LW heating near 100 hPa of about 0.1 to 0.2 K/day due to subvisible cirrus, in particular during night, when the CALIPSO lidar better detects subvisible cirrus. The small SW clear

sky heating positive bias of the same order of magnitude between 400 and 800 hPa is most probably linked to contamination by partial cloudiness.

Relatively opaque clouds heat the atmospheric column below by trapping surface emissions, but cool the column above due to excess emission, while thin cirrus heat the UT by intercepting the LW radiation coming from below. Indeed, Figure 3 exhibits a LW cooling above optically thick clouds, the strongest effect above Cb, of about -4.5 K/day around 170 hPa, and a

heating within the clouds and below the clouds, compared to clear sky. The small cooling around 550 hPa is due to melting. The cooling above mid- and low-level clouds is located around 600 hPa and 800 hPa, respectively. Thin cirrus heat the UT around 100 hPa. In the SW range, the sunlight heats the atmosphere and the particles within the cloud. We observe a strong heating in the upper part of the Cb with a maximum of about 8 K/day around 200 hPa, while in the rest of the cloud this effect is negligible, given that the sun is blocked by the dense cloud particles. For midlevel clouds a small peak is found around 600

hPa and for low-level clouds around 850 hPa. The LW HRs are very similar during day and night. Although LW and SW contributions are large, their effect is opposite so that the net radiative heating during day (LW + SW) is small. This agrees very well with the expectations and earlier publications (e.g. Oreopoulos et al., 2016).

To illustrate the additional value of the lateral expansion of the radiative HRs, Figure 4 presents geographical maps of monthly mean LW heating / cooling in four specific pressure layers (around 106, 200, 525 and 850 hPa, respectively) compared to the

monthly mean nadir track statistics from CALIPSO-CloudSat. These four pressure layers were chosen according to 1) UT heating by thin cirrus, 2) cooling above Cb and thick cirrus, 3) middle troposphere heating by high thick clouds and 4) cooling above low-level clouds and a heating below clouds. The horizontal structures of the predicted HR fields agree quite well with



those from FLXHR, but they appear clearer, since the spare nadir track statistics is quite noisy. The comparison between the fields determined from different models in Figure S6 of the supplement gives an indication of the uncertainty, which lies
generally within 0.25 K/day, with a few regions of 0.45 K/day.

## 4 The impact of tropical UT cloud systems

By using the 3D radiative heating fields constructed in section 3, we first want to quantify the effect of clouds on the atmospheric radiative cooling, in comparison to earlier results (section 4.1). Furthermore the cloud system approach described in section 2.1 allows us to study the heating and cooling within convective cloud systems by distinguishing the HRs of the
convective cores (Cb), the cirrus anvil (Ci) and the surrounding thin cirrus (thin Ci). This may yield a deeper insight how the tropical heating changes with varying surface temperature (sections 4.2 and 4.3).

### 4.1 Tropics-wide cloud radiative heating

In general, clouds introduce sharp vertical gradients to the atmospheric radiative cooling profile, and we are in particular interested in the effect of MCSs. Li et al. (2013) have concluded that the tropics-wide 24-hr mean UT cloud radiative heating
effect has a narrow maximum of about 0.45 K/day around 250 hPa, and that the column-integrated radiative heating of UT clouds accounts for about 20% of the latent heating estimated by TRMM, the latter with a broad peak of about 1.7 K/day around 450 hPa. These results were obtained by using radiative heating rates calculated from ground-based lidar and radar measurements at two ARM sites (Manus and Darwin), classified by ISCCP UT cloud regimes, and then expanded over the deep tropics according to the ISCCP UT cloud regime occurrence frequency.
In order to compare to this significant result, we concentrate on the same latitude band from 15N to 15S and we calculate the 24-hr SW heating rates by multiplying the SW heating rates at 1:30PM LT by $1/(\pi \times \cos\Theta)$, where $\Theta$ is the solar zenith angle. The latter is about 33° near the equator. Similar to the HR normalisation of Li et al. (2013), we neglect seasonal and geographical variations. The net radiative heating is then the sum of LW and 24 hr SW heating. This estimation assumes a negligible diurnal cycle in clouds. Indeed, the diurnal cycle of UT clouds over tropical ocean is less than 2% and reaches about
7% over tropical land (Feofilov and Stubenrauch, 2019), with slightly less cirrus and thin cirrus at 1:30PM than at 1:30AM. This means that our estimated 24 hr net heating effect of UT clouds, thin cirrus and MCSs is slightly underestimated.
We determined the average 24 hr net radiative heating effect of specific cloud types by removing the clear sky radiative heating rate and then weighted by the total cloud cover. Figure 5 presents a tropics-wide 24 hr ó mean radiative heating induced by UT clouds throughout the troposphere, from 250 hPa downward, of about 0.3 K/day, with a broad maximum of about 0.4
K/day around 330 hPa and a local minimum of about 0.2 K/day around 550 hPa due to ice crystal melting at this altitude, already reported by Johannsson et al. (2015). The heating decreases towards 0 K/day at 200 hPa, and above this altitude a



small net cooling is observed. The values are in the same range as the ones determined by Li et al. (2013). However, the shape of the heating effects are significantly different: Whereas the earlier result shows a narrow maximum of 0.45 K/day around 250 hPa and a minimum heating of about 0.1 K/day around 800 hPa, our estimation indicates a much more vertically extended

heating, with a broader maximum of about 0.4 K/day around 330 hPa and a heating throughout the lower troposphere larger than 0.25 K/day. This then leads to a vertically different reinforcement of the latent heating, with a larger contribution between 800 hPa and 330 hPa, compared to Figure 9 of Li et al. (2013), shown in Figure S7 in the supplement. The enhancement factor between our column-integrated radiative heating of UT clouds and this latent heating (between 100 and 800 hPa) is 24%, larger than 21% found by Li et al. (2013).

The difference in the profile shape of the UT cloud radiative heating effect is not related to the exploitation of profiles from only two sites, as the profiles of the different cloud types, when present, averaged over both sites are similar to the ones averaged over the whole tropics (not shown). However, as discussed by Protat et al. (2014), a significant portion of the ice cloud observations using ground-based measurements is attenuated by any liquid cloud below ice clouds or by the liquid part of deep convective systems. This leads to a smaller SW heating than the satellite estimates in the middle troposphere. Another

key reason for an underestimation of the heating effect in the lower troposphere is that the ISCCP cloud regimes have been determined at a spatial resolution of 2.5° and especially the cirrus and mixed cloud regimes, which are the most frequent out of the four UT cloud regimes (72%), include also a certain fraction of single-layer low-level clouds next to the cirrus clouds. When considering the radiative effect of mid- and lowlevel clouds in Figure 5, which shows a cooling in the middle and lower troposphere down to 880 hPa, the shape of the radiative heating profile contribution of the ISCCP UT cloud regimes can be

explained by the fact that at the coarse spatial resolution of 2.5° the UT cloud regimes also contain surrounding single-layer low-level clouds. In addition, the identification of thin cirrus with optical depth less than 1.3, the most frequent within these two ISCCP cloud regimes, is also less reliable, and the cloud height in this case is often just set to the tropopause height (e. g. Stubenrauch et al., 2012).

Further consideration of Figure 5 reveals that most of the total cloud net radiative heating effect, which is the sum of the UT

and mid- and low-level cloud effects, comes from MCSs. The UT cooling above the opaque parts of the MCSs is compensated by thin cirrus UT heating, with half of the effect coming from those directly surrounding the anvil and the other half from in situ cirrus. The average net radiative heating within and the cooling above the MCSs seems to be slightly stronger over ocean than over land. Mid- and low-level clouds present a cooling above the clouds and a heating within and below. As there are more low-level clouds over ocean and more mid-level clouds over land, the shapes of the net heating effect differ accordingly.

The HR profiles of UT clouds, initially deduced from CALIPSO-CloudSat data, include the effect of lower clouds underneath, as the small warming peak around 920 hPa suggests. This peak is slightly stronger over ocean, where one expects more low-level clouds, also underneath cirrus.





From Figure 6, which compares the tropics-wide mean net radiative heating effect of the different cloud types at 1:30 AM LT and at 1:30 PM LT, we deduce a large difference in the profile shapes between nighttime and daytime, and therefore in their
vertical heating gradients. During nighttime, UT clouds heat the troposphere from 300 hPa downward increasingly, with a maximum of about 0.6 K/day around 920 hPa. The thicker UT clouds lead to an average cooling, with a minimum of -0.25 K/day around 200 hPa. This leads to a strong vertical gradient. The heating of the lower troposphere is slightly larger over land, but with a smaller vertical gradient in the lower troposphere. Thin cirrus have a very small average heating effect around 150 hPa, slightly larger over land than over ocean. During daytime, with additional solar heating, UT clouds, in particular the
thicker ones, are strongly heated (see also Figure 3), which leads to a tropics-wide maximum of about 0.6 K/day between 250 and 350 hPa. The heating strongly decreases towards the lower troposphere. Again, most of the effect of UT clouds can be explained by MCSs. The profiles reveal slightly lower but stronger systems over ocean and slightly higher systems over land, with the peak in heating located about 100 hPa higher in altitude over land than over ocean.

During nighttime and during daytime, thin cirrus have on average a small heating effect throughout the whole troposphere.
The effect of low-level and midlevel clouds differs diurnally: During nighttime we observe a cooling effect above these clouds, of -0.3 K/day around 820 hPa and of -0.1 K/day around 550 hPa, respectively, and a heating within and below, while during daytime the SW contribution partly compensates this effect. In general, the UT cloud effect is a strong heating of the UT during daytime and a strong lower tropospheric heating during nighttime, leading to opposite vertical gradients.

### 4.2 Relation between regional surface temperature and MCSs

A necessary condition for the onset of tropical deep convection, particularly over ocean, is a surface temperature ($T_{surf}$) above a threshold of about 300 K (e. g. Gray, 1968; Graham and Barnett, 1987, Aumann et al., 2018), though other factors, such as available humidity (which may increase with low-level level convergence), also affect the convective process. Though the shading of the thick anvils may cause some surface cooling during day, slightly offset by the thinner cirrus (Wall et al., 2018), MCSs should be deeper over warm regions (here defined by $T_{surf} > 302$ K) than over cool regions ($T_{surf} < 300$ K). As in a
changing climate the extension of warm regions may slightly increase, we compare in this section the properties of MCSs over warmer and over cooler regions.

Figure 7 presents geographical maps of UT cloud occurrence, average precipitable water and surface temperature, separately during nighttime (1:30 AM LT) and during daytime (1:30 PM LT). MCSs are most frequent over the West Pacific ocean, including Indonesia, over the Amazon region and over Central Africa. These are also the moistest regions. The West Pacific
region is also characterized by the warmest $T_{surf}$ values. While over the oceanic region we do not observe a diurnal difference in UT cloud occurrence, over the two land regions UT clouds are more frequent during night, as convection starts in the early evening and cirrus anvils develop during night.





In Figure 8, distributions of the properties of maritime and continental MCSs are then compared between the warm regions and cool regions. Since the size of the MCSs may be very large (Protopapadaki et al., 2017), we only use the 50% warmest underlying grid cells to determine the mean $T_{surf}$. In general, these warm regions are also more humid according to the distributions of precipitable water (from ERA-Interim), also shown in Figure 8. As expected, both, maritime and continental MCSs overlying warm regions have colder convective cores, which means that their convective depth is larger, and also have more often a larger horizontal extent, in agreement with a regional study by Horvath and Soden (2008), than those overlying cool regions. The area occupied by the thin cirrus surrounding their anvils is also larger for the first case. This can be explained by i) a larger relative humidity at higher altitude and ii) additional UT humidification originating from the convection.

The tropics-wide 24-hr mean net radiative heating effect of the MCSs depends on their frequency, height, horizontal extent and emissivity structure. In order to study the first aspect, Figure 9 contrasts the effect of mid- / low-level clouds and UT clouds, when these are present, and also shows the resulting overall cloud effect, over cool and warm regions, respectively. Furthermore, the effects of MCSs, of the thin cirrus linked to the MCSs and of all thin cirrus are distinguished. First of all, over warm regions, clouds, when present, have a heating effect over most of the troposphere, and this heating is mostly driven by MCSs. This can be seen from the fact that the profiles of the present MCSs and those of all clouds are very similar. Over cool regions mid- and in particular low-level clouds (over ocean) also play an important role, with much less heating between 200 and 900 hPa than the one of UT clouds. Over warm regions, the UT thin cirrus heating linked to convection is larger than the one of all thin cirrus, which indicates more and slightly thicker thin cirrus linked to convection than those produced in situ.

The influence of emissivity structure is investigated by considering the 24-hr mean net heating / cooling effects of the different parts of the MCSs, convective core (Cb), Cirrus anvil (Ci) and surrounding thin Cirrus (thin Ci), when MCSs are present. These are shown in Figure 10, for all tropical MCSs, separately over ocean and over land, and those over cool and warm regions, respectively. As already seen in Figure 3, the shape of the vertical profiles is quite different for the three parts of the MCSs. In the UT (at a height above 200 hPa), we observe an average cooling of about -2 K/day above the convective cores and a less strong cooling above the cirrus anvil, while the thin cirrus heat the UT by about 0.7 K/day. The troposphere below the height of 200 hPa is strongly heated by the convective cores, much less heated by the cirrus anvils and even less by the surrounding thin cirrus. However, as the convective cores only cover a small fraction of the systems (about 10% on average), the average heating effect of the MCSs is about the one of the cirrus anvils. On average, the profiles of the three parts of the MCSs have a similar shape over ocean and over land, but the net effect of the MCSs over ocean is slightly larger than over land, because the convective core fraction is smaller and the thin cirrus proportion is larger for the latter. The difference in shape of the heating profiles between cool and warm regions is larger for oceanic MCSs. For the latter, the shape of the heating profiles strengthens the hypothesis of MCSs with larger convective depth above the warm regions, with a cooling of the thicker parts of the MCSs shifted further up into the UT by 50 hPa, while the heating is extended over a broader vertical layer between





550 to 200 hPa. On the other hand, over both, ocean and land, the thin cirrus net radiative heating of the UT of about 0.6 K/day

is only connected to the deeper convective systems over the warm regions.

Gasparini et al. (2019) found that the vertical structure of the radiative heating within the anvil promotes its spreading and maintenance. The spreading then influences the heating gradients within the MCSs. To illustrate probable differences in horizontal gradients within the MCSs over cool and warm regions, we present in Figure 11 the difference of mean net radiative HR between cirrus anvil and convective cores as well as between surrounding thin cirrus and cirrus anvil, both divided by the

distance between these parts, assuming a system with a circular surface. This is only a very rough estimate, and to test the robustness we present averages over all MCSs and over those which have more probably a circular surface: MCSs with only one convective core and a core fraction larger than 0.05. In both cases, the ¬horizontal heating / cooling gradientsø within the MCSs seem to be slightly smaller in the warm regions than in the cool regions. The effects are stronger for the confirmed single core MCSs, which corroborates the quality of our data. While the horizontal effect is slightly larger over land, over

ocean we also observe a slight vertical shift in the horizontal gradients (Figure S12). The effect is larger in the SW and therefore larger during daytime than during nighttime (Figure S13). This difference in ¬horizontal gradientøcan be solely understood by the larger anvil size and therefore larger distance between the average HRs between the different parts of the MCSs, and not by a larger emissivity of the anvil and the surrounding thin cirrus, as on the contrary the average anvil emissivity is slightly smaller in the warm regions than in the cold regions (Figure S14). The latter is in agreement with a study of Del Genio et al.

(2005), which revealed a decreasing detrainment and increasing precipitation efficiency within maritime MCSs when the underlying $T_{surf}$ increases.

### 4.3 Changes in tropical heating and in MCSs in dependence of global surface temperature anomaly

In section 4.2 we have shown that the heating in the warmer tropical regions is mostly influenced by MCSs. In this section we look at variations in MCSs and in tropical heating and try to relate these to global $T_{surf}$ anomalies and to phenomena which

influence the interannual variability. The time period covered by AIRS observations may still not be long enough for climate change attribution, and in particular during most of this period global $T_{surf}$ was quite stable. However, when comparing to linearly increasing $CO_2$ concentration, we perceive the $T_{surf}$ anomalies undulating around and thus slightly increasing (Figure 12). A large attribution to the global $T_{surf}$ anomalies, which are closely associated with tropical $T_{surf}$ anomalies (see Figure S14 of the supplement), is given by the El-Niño-Southern Oscillation (ENSO), the most dominant mode of interannual variability

in the Earthøs climate system. El Niño (La Niña) events are linked to a positive (negative) tropical $T_{surf}$ anomaly. Their initiation is given by a local $T_{surf}$ anomaly (positive $T_{surf}$ anomaly in the equatorial eastern and central Pacific or negative $T_{surf}$ anomaly in the tropical Pacific), changing the east-west $T_{surf}$ gradient, which then affects the atmospheric circulation (reducing or amplifying the Walker circulation) and the distribution of clouds. However, the magnitude and also to some extent the geographical pattern of El Niño induced $T_{surf}$ anomalies contribute to differences in cloud and circulation anomalies (e. g. Su



and Jiang, 2013). The Inter-decadal Pacific Oscillation (PDO), another $T_{surf}$ anomaly in the Pacific, can be influenced by ENSO. The combination of both indices, the Oceanic Niño Index and the NCEI PDO index, both provided by the National Oceanic and Atmospheric Administration (NOAA), are therefore also presented in Figure 12.

The average coverage of all UT cloud systems over the period 2003 to 2018 is 25.6%, with 6% from thin cirrus systems and 80% of their coverage from MCSs. 48% of the latter are cold MCSs with T < 210 K. In order to estimate the changes in the

properties of the tropical MCSs in relation with global $T_{surf}$ change, we determined the linear regression slopes between the anomalies of the MCS properties and the global $T_{surf}$ anomalies. While the coverage of the MCSs is relatively stable (or very slightly decreasing) with warming, with -1.3 ± 0.6 %/K, the coverage of cold MCSs relative to all MCSs significantly increases by +18 ± 5 %/K. The latter can also be expressed by a decreasing cloud system temperature of ó 2.1 + 0.5 K/K. Furthermore, the surrounding thin cirrus area relative to the anvil area increases slightly by +0.041 ± 0.008 / K. The time series of 12-month

running means of the anomalies of the coverage of cold MCSs relative to all MCSs, of the minimum convective core temperature ($T_{Cbmin}$) and of the area of surrounding thin cirrus relative to cirrus anvil area are also shown in Figure 12. Indeed, they are related to global $T_{surf}$ anomalies as well as to ENSO.

When considering the time series of the anomalies of the vertical heating / cooling effect of the MCSs in Figure 13, we observe vertical dipole effects in the net during daytime, which seem to be strongly linked to ENSO variability and can be explained

by changes in convective depth of the MCSs. The anomalies have values of about -0.4 and +0.4 K/day, respectively. During La Niña periods we observe a cooling anomaly in the UT above the height corresponding to 350 hPa and a heating anomaly in the atmosphere below, which suggests less strong MCSs. During El Niño periods, the HR vertical structure anomaly seems to suggest deeper MCSs, which moves the cooling above the thick anvils and the heating within and below upward. In the LW, we observe a pattern of heating and cooling anomalies above the height corresponding to 200 hPa, in accordance with ENSO,

with more thin cirrus heating during La Niña and more thick anvil cooling during El Niño. It is interesting to note that the anomalies are slightly larger during nighttime.

Zelinka and Hartmann (2010) found during El Niño periods a decrease of high-level cloud amount as well as an increase in their height which would have opposite effects on the OLR, with a dominating effect coming from the first. Therefore we also investigate the time series of heating / cooling anomalies for all clouds together in Figure 14. The anomalies in the upper and

middle troposphere have the same pattern as the ones for the MCSs, only much smaller, because their relative frequency of occurrence is taken into account. In addition, we observe strong LW cooling and heating anomaly patterns in the lower atmosphere, linked to the occurrence of stratocumulus and stratus cloud fields. It looks like there are more extended or thicker stratocumulus / stratus cloud fields when the convective systems are deeper within the latitudinal band and there are less or less thick when the convective systems are less deep. This confirms that height and extent of MCSs on one hand and extent of

the stratocumulus and stratus fields on the other are energetically constrained within the tropics and subtropics (e.g. Hang et al., 2019; Jakob et al., 2019).





## 5 Conclusions and Outlook

The active lidar and radar measurements from CALIPSO and CloudSat make it possible to derive radiative HR profiles, but only on narrow nadir tracks. On the other hand, AIRS, also part of the A-Train satellite constellation, provides cloud properties

with a large instantaneous horizontal coverage. By using these radiative HRs for the training and applying the resulting ANN models on cloud properties from AIRS and atmospheric and surface properties from ECMWF meteorological reanalyses, we constructed 3D HR fields within 30N to 30S, for the period 2003 to 2018.

We demonstrated that non-linear ANN regression models, trained on four years of collocated data along the nadir tracks, are appropriate methods to estimate tropical radiative HRs from about 40 cloud, atmospheric and surface properties. Column-

integrated MAE is about 0.8 K/day (0.5 K/day) for cloudy scenes and 0.4 K/day (0.3 K/day) for clear sky in the LW (SW). When evaluating the profile shapes, developing separate models for i) Cb, ii) cirrus and thin cirrus, iii) mid- and low-level clouds and iv) clear sky, independently over ocean and over land, leads to a small improvement, with the mean differences between predicted and ÷observedørradiative HRs well undulating around 0 K/day. The improvement is most noticeable for Cb, with uncertainties around the maxima of LW cooling and SW heating linked to slight vertical shifts between the different

models. The monthly mean horizontal structures of the predicted HR fields agree well with the original ones from CALIPSO-CloudSat (2B-FLXHR-LIDAR R04), but they are more obvious, due to the lateral expansion. The comparison between the fields determined from different models gives an indication of the uncertainty, which lies generally within 0.25 K/day per layer.

The input variable normalization using maximum and minimum guarantees that the regression models produce also reliable

results outside the training period (assuming a non-changing relationship between the input parameters and the HRs), made clear by the long-term temporal behaviour of the HRs, in particular in relation to ENSO variability.

We confirmed that most of the total cloud net radiative heating effect in the deep tropics (15N-15S) comes from UT clouds. These have a 24hr mean net radiative heating effect larger than 0.25 K/day throughout the troposphere from 250 hPa downward, with a broad maximum of about 0.4 K/day around 330 hPa, enhancing the column-integrated latent heating

(between 100 and 800 hPa) by 24%. This value is larger than earlier results of about 20%, using ISCCP cloud data. Our result may still be slightly underestimated, because of the not comprised diurnal variation of UT clouds, the cloud contamination of the clear sky scenes identified by AIRS and the slightly underestimated LW warming above 12 km in the original FLXHR-LIDAR (R04) data linked to cirrus microphysical assumptions. However, the shape of the heating effects compared to those of Li et al. (2013) are significantly different, with our estimation indicating a much more vertically extended heating. This

suggests an underestimation of the heating in the middle troposphere of the earlier result, which can be explained by the shading effect of underlying low-level clouds on ground-based measurements and by a mixture of cirrus and surrounding single-layer low-level clouds linked to the coarse spatial resolution.





Above the height corresponding to 200 hPa, the LW cooling above the opaque parts of the MCSs is compensated by thin cirrus heating, with half of the effect coming from those directly surrounding the anvil and the other half from in situ cirrus. In
general, the UT cloud effect is a strong heating of the UT during daytime and a strong lower tropospheric heating during nighttime, leading to opposite vertical gradients. The heating profiles also reveal slightly lower but stronger systems over ocean and slightly higher systems over land, with the peak in heating located about 100 hPa higher in altitude over land than over ocean.

The shapes of the heating profiles for the three parts of the MCSs (convective cores, cirrus anvil and surrounding thin cirrus)
differ significantly. The troposphere below the height of 200 hPa is strongly heated by the convective cores, much less heated by the cirrus anvils and even less by the surrounding thin cirrus. However, as the convective cores only cover a small fraction of the systems, the average heating effect of the MCSs is about the one of the cirrus anvils.

Mid- and low-level clouds produce a cooling above the clouds and a heating within and below. As there are more low-level clouds over ocean and more mid-level clouds over land, the shapes of the net heating effect differs accordingly.

MCSs are most frequent over the West Pacific, including Indonesia, over the Amazon region and over Central Africa. These are also the moistest regions. As expected, both, maritime and continental MCSs overlying warmer regions have colder convective cores, which means that their convective depth is larger, and the area occupied by the thin cirrus surrounding their anvils is also larger. The latter phenomenon can be explained by i) larger relative humidity at higher altitude and ii) additional UT humidification originating from the convection.

Over warm regions ($T_{surf} > 302$ K), the heating is mostly driven by MCSs, which also have a larger convective depth than the ones over cool regions ($T_{surf} < 300$ K). The consequence is a heating over a broader vertical layer, between 550 to 200 hPa. The thin cirrus linked to the MCSs in these regions heat the UT by about 0.7 K/day, more than the in situ formed cirrus. The latter play a more important role over cool regions, as well as mid- and low-level clouds (over ocean), with much less heating between 200 and 900 hPa.

The anvil spreading influences the heating gradients within the MCSs. A rough estimate of the ‒horizontal heating / cooling gradientsø within the MCSs exhibits slightly smaller ‒horizontal gradientsø in the warm regions than in the cool regions. The effect is larger in the SW and therefore larger during daytime than during nighttime. This difference in ‒horizontal gradientø can be solely understood by the larger anvil size.

During the time period 2003 to 2018, a large attribution to the global $T_{surf}$ anomalies is given by ENSO, with El Niño (La Niña)
events linked to a positive (negative) $T_{surf}$ anomaly. The time series of the anomalies of the vertical heating / cooling effect of the MCSs exhibits vertical dipole effects in the net during daytime, strongly related to ENSO variability and explained by changes in convective depth of the MCSs: During El Niño periods, the HR vertical structure anomaly suggests deeper MCS, with vertically broader heating. The LW heating and cooling anomalies above the height of 200 hPa is also in accordance with ENSO, with more thin cirrus heating during La Niña and more thick anvil cooling during El Niño. The cloud heating / cooling





anomaly patterns in the upper and middle troposphere are dominated by the MCSs, and the LW cooling / heating anomaly patterns in the lower atmosphere exhibit more stratocumulus cloud fields when there are more deep convective systems within the latitudinal band and less stratocumulus cloud fields when the MCSs are less deep. This confirms that height and amount of UT clouds on one hand and extent of the stratocumulus fields on the other are energetically constrained within the tropics and subtropics.

With respect to global $T_{surf}$ anomalies, the MCS coverage is relatively stable (or very slightly decreasing) with warming, with -1.3 ± 0.6 %/K, while the coverage of cold MCSs relative to all MCSs significantly increases by +18 ± 5 %/K. Furthermore, the surrounding thin cirrus area relative to the anvil area increases slightly by +0.041 ± 0.008 / K.

In the future, we will add the latent heating profiles derived from the Tropical Rainfall Measuring Mission (TRMM) to this synergistic data set, which provides for the first time a 3D view of the radiative heating profiles over a long time period,. As 645 the coincidences in time with AIRS are small, we will use again machine learning techniques, similar to the ones described in this article. This data base of UT cloud systems is being constructed within the framework of the GEWEX (Global Energy and Water Exchanges) Process Evaluation Study on Upper Tropospheric Clouds and Convection (GEWEX UTCC PROES, https://gewex-utcc-proes.aeris-data.fr/) to advance our knowledge on the climate feedbacks of UT clouds. In general, climate feedback studies are undertaken by climate model simulations, which rely upon their representation of convection and 650 detrainment. The cloud system approach has already proved its usefulness in the evaluation of a new bulk ice cloud scheme in the LMD GCM (Stubenrauch et al., 2019), and the HRs may be used to distinguish between parameterizations of ice cloud radiative properties. Furthermore, this data base, in particular when including the total 3D diabatic heating, will be used to quantify the dynamical response of the climate system to the atmospheric heating induced by the anvil cirrus, refining and extending the studies of Schumacher et al.(2004) and Li et al. (2013).

**6 Acknowledgements and Data**

This work is supported by the Centre National de la Recherche Scientifique (CNRS) and the Centre National d¢Etudes Spatiales (CNES). The authors thank the members of the AIRS, CALIPSO and CloudSat science teams for their efforts and cooperation in providing the data, as well as the engineers and space agencies who control the data quality. AIRS CIRS data have been produced by the French Data Centre AERIS and will be distributed at https://cirs.aeris-data.fr. In section 4.3, we used global 660 $T_{surf}$ anomalies derived by the GISTEMP Team, (2020). The dataset was accessed in 2019 at https:// data.giss.nasa.gov/gistemp/. Monthly indices of ENSO strength (ONI) and of PDO were obtained from NOAA (https://origin.cpc.ncep.noaa.gov/products/analysis_monitoring/ensostuff/detrend.nino34.ascii.txt and https://www.ncdc. noaa.gov/teleconnections/pdo/, respectively).





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

**Figures**

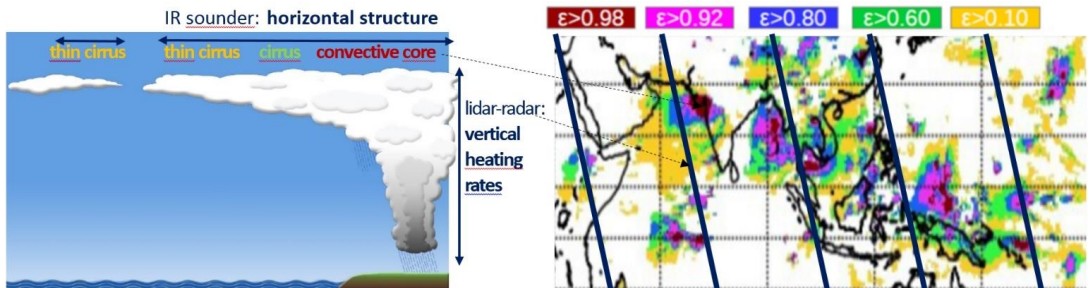

**Figure 1: Illustration of three dimensional Cloud System Concept, using spaceborne IR Sounder data (AIRS), providing the horizontal component, and lidar-radar data (CALIPSO-CloudSat), providing the vertical component, both part of NASA's A-Train**

**satellite constellation (left): Based on two independent variables retrieved by AIRS, UT cloud systems are reconstructed from adjacent elements of similar cloud height ($p_{cld}$), the horizontal emissivity structure allows to directly link the properties of convective cores ($\varepsilon_{cld} > 0.98$) and cirrus anvils (right). Clear sky and low-level cloud fields are also identified (Fig. 4a of Protopapadaki et al. 2017). A horizontally complete picture of the vertical radiative heating rates will be obtained by laterally expanding them, as they are only available along narrow lidar-radar tracks (dark blue). Therefore we have developed optimized "non-linear regression**

**models", using deep neural network learning techniques, described in section 2.5 and evaluated in section 3, to relate the most suitable cloud and atmospheric properties from IR sounder and meteorological reanalyses to these heating rates.**

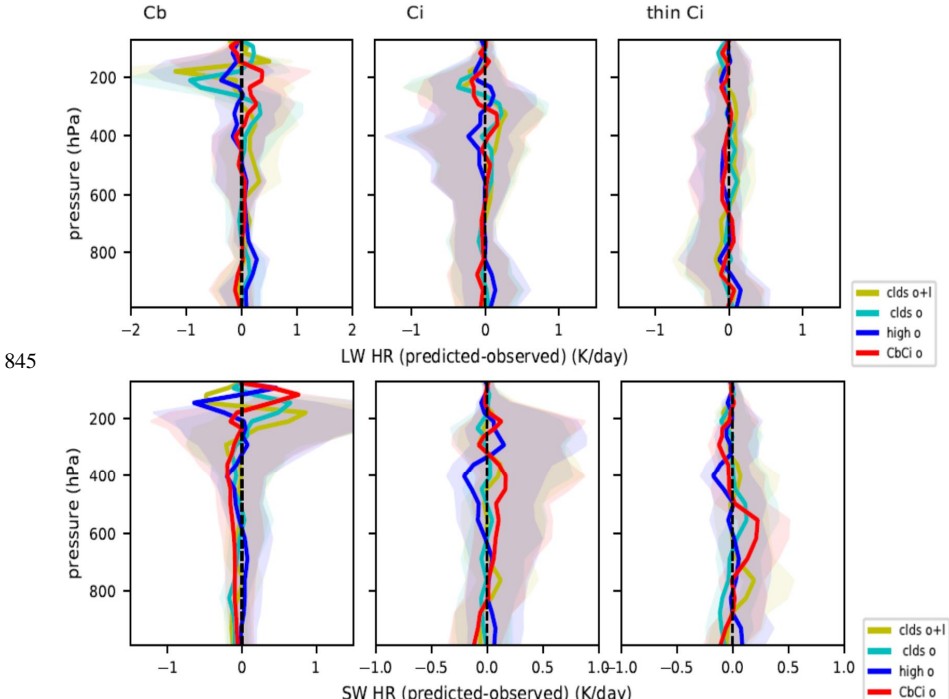


**Figure 2: Sensitivity results concerning training over different scenes (high clouds over ocean, all clouds over ocean and all clouds over ocean and land) for the prediction of high-level cloud LW radiative heating rates (above) and SW radiative heating rates (below): difference between predicted and observed vertical profiles, separately for Cb, Cirrus and thin Cirrus, as identified by**
**AIRS-CIRS, over tropical ocean. 30% and 70% quantiles of the distributions are also shown.**

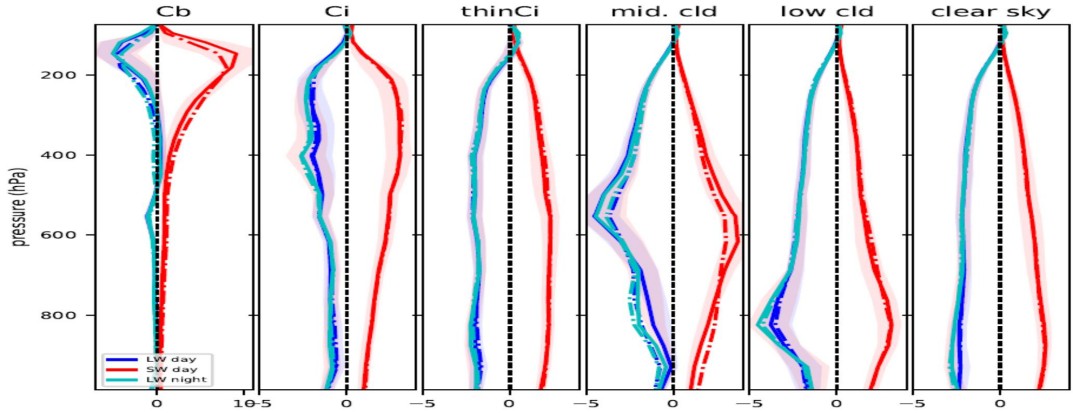

**Figure 3: Predicted LW heating rates and SW heating rates (full line), separately for Cb, Cirrus, thin Cirrus, mid-and low-level clouds and clear sky, as identified by AIRS-CIRS, averaged over the AIRS swaths within 30N ó 30S, in January 2008. 30% and 70% quantiles of the distributions indicate their variability. The model has been trained individually over Cb, Ci / thin Ci and mid- / low-level clouds, separately over ocean and land. Broken lines correspond to the average of FLXHR heating rates averaged along the CALIPSO-CloudSat nadir tracks. Night corresponds to 1:30 AM and day to 1:30 PM local time.**

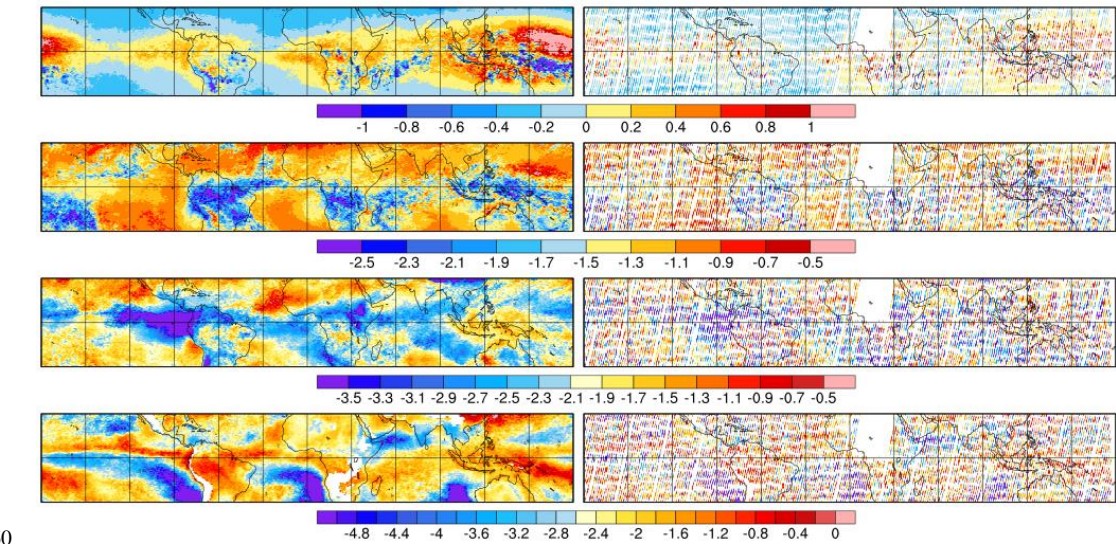


**Figure 4: Geographical maps of LW heating rates (K/day) in 4 layers: 106-131 hPa, 200-223 hPa, 525 ó 585 hPa and 850 ó 900 hPa (from top to bottom) averaged over January 2008 at 1:30AM. Left: predicted over the AIRS swath, using the combination of the eight models developed for Cb, Ci / thin Ci, mid- / low-level clouds and clear sky, separately over ocean and over land. Right: from**
**NASA FLXHR data along the CALIPSO-CloudSat nadir tracks.**





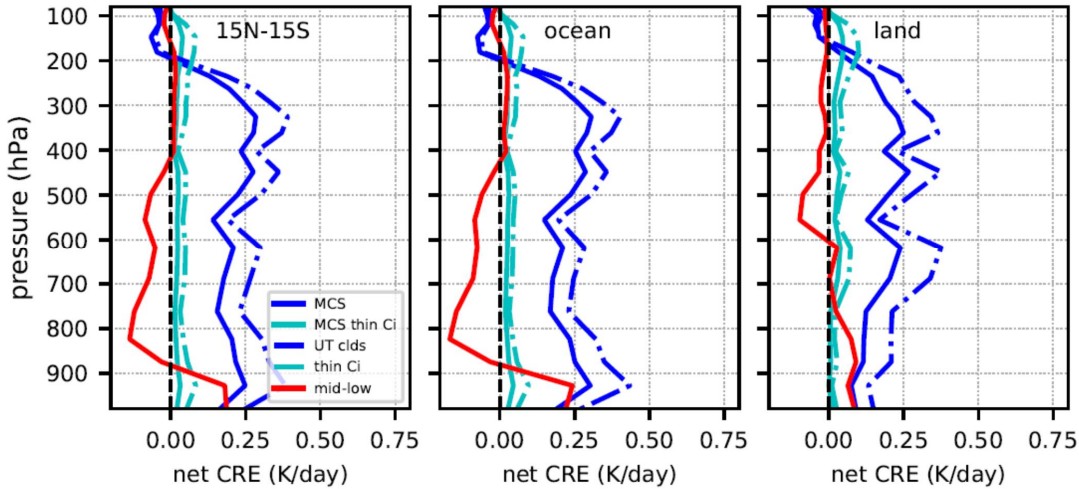

**Figure 5: Tropical mean net radiative heating effect within the troposphere of low- and mid-level clouds (red) and UT clouds (blue, broken line), for the latter the effect of MCSs (blue), thin cirrus surrounding MCSs (cyan, full line) and all thin cirrus (cyan, broken line) is shown separately. Left: all, middle: ocean, right: land. Cloud observations at 1:30PM local time, with SW radiation normalized to 24 hours, similar to Li et al. (2013). Statistics of 15 years (2004 ó 2018), averaged over 15N to 15S. The sum of UT cloud and mid- / lowlevel cloud contributions corresponds to the total cloud heating effect, defined as the difference between total and clear sky heating.**



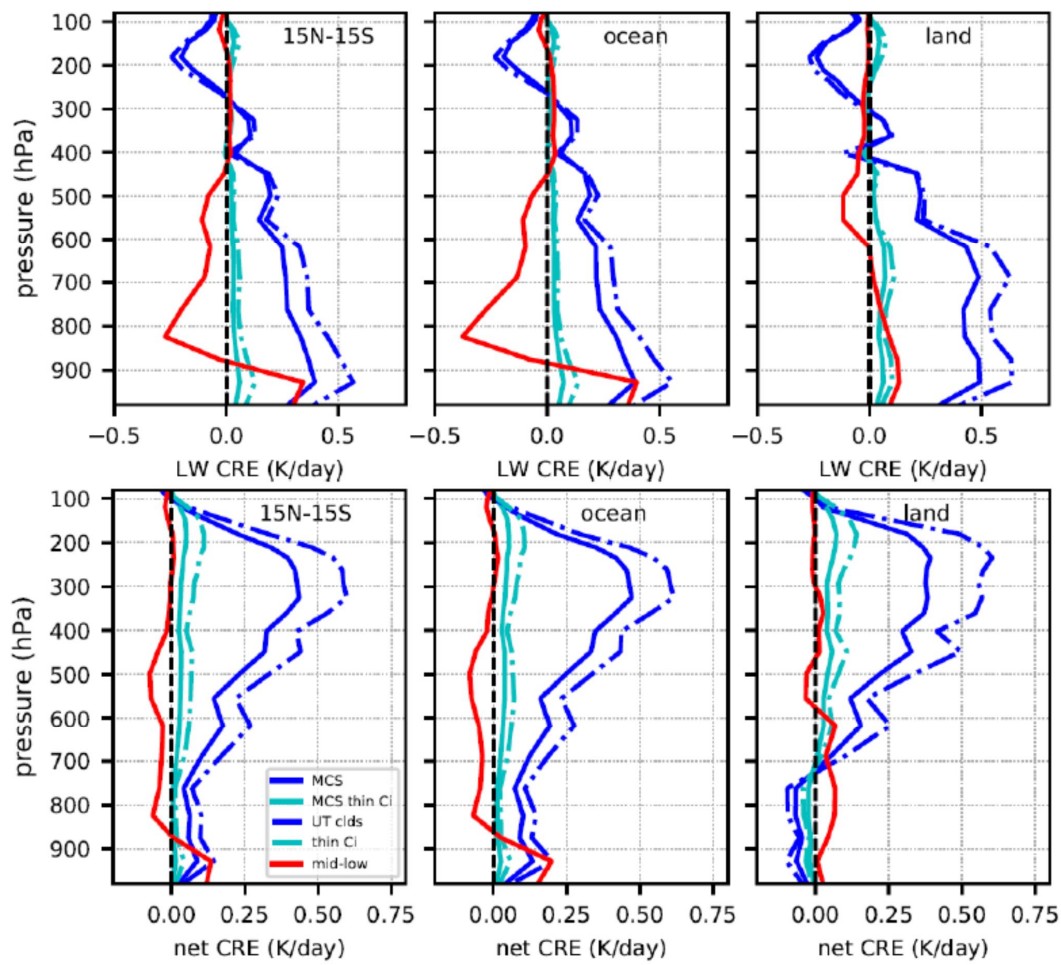

**Figure 6: Tropical mean net radiative heating effect within the troposphere of low- and mid-level clouds (red) and UT clouds (blue, broken line), for the latter the effect of MCSs (blue), thin cirrus surrounding MCSs (cyan, full line) and all thin cirrus (cyan, broken line) is shown separately. Left: all, middle: ocean, right: land. Above: at 1:30AM local time, below: at 1:30PM local time. Statistics of 15 years (2004 ó 2018), averaged over 15N to 15S.**



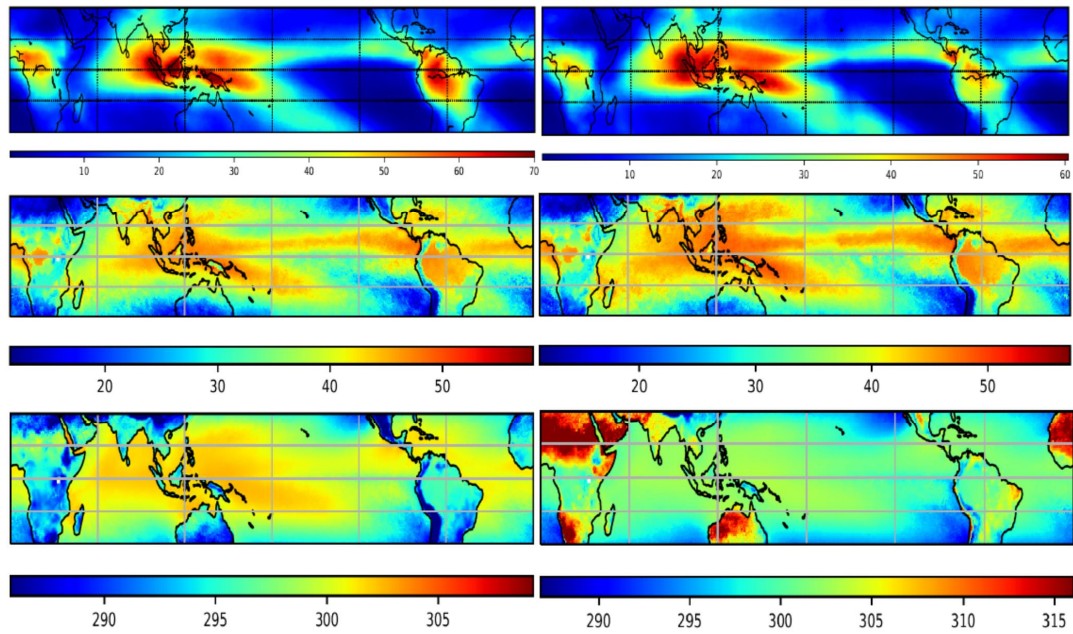

**Figure 7: Geographical maps of occurrence frequency of UT clouds (top row), of total precipitable water in mm (middle row) and of surface temperature in K (bottom row), at 1:30 AM LT (left) and at 1:30 PM LT (right). Statistics of 16 years (2003-2018).**




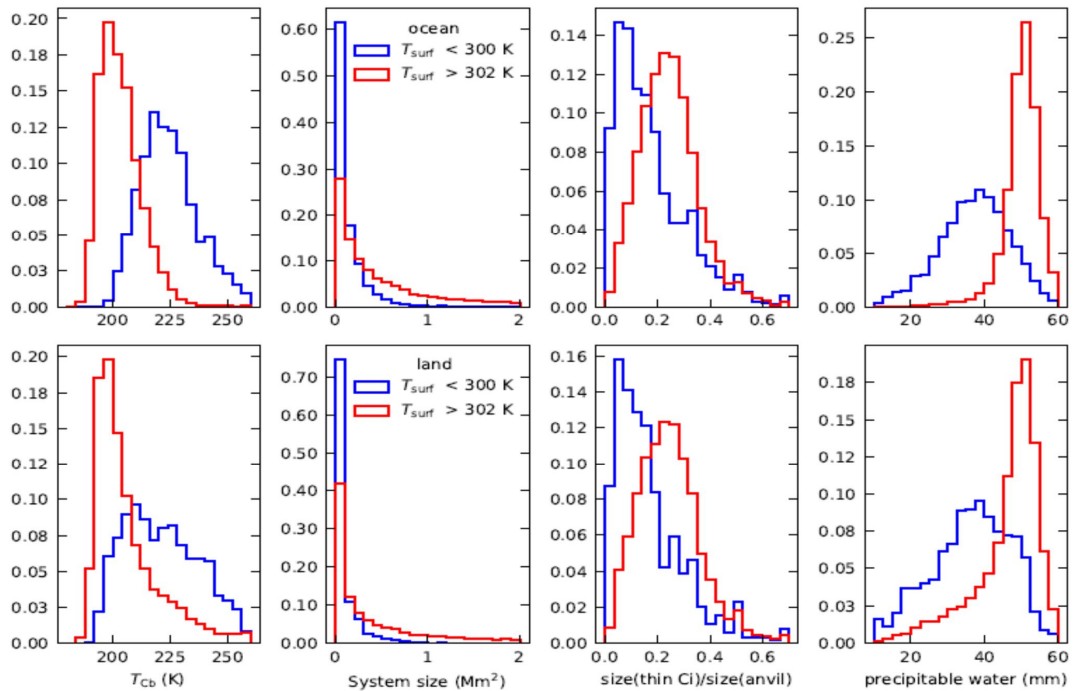

**Figure 8: Normalized distributions of properties of the tropical MCSs (temperature of convective core, cloud system size, relative size of the thin cirrus within or surrounding the anvil) and atmospheric precipitable water, separately for systems with a smaller and a larger underlying surface temperature, over ocean (above) and over land (below), at 1:30 PM LT.**

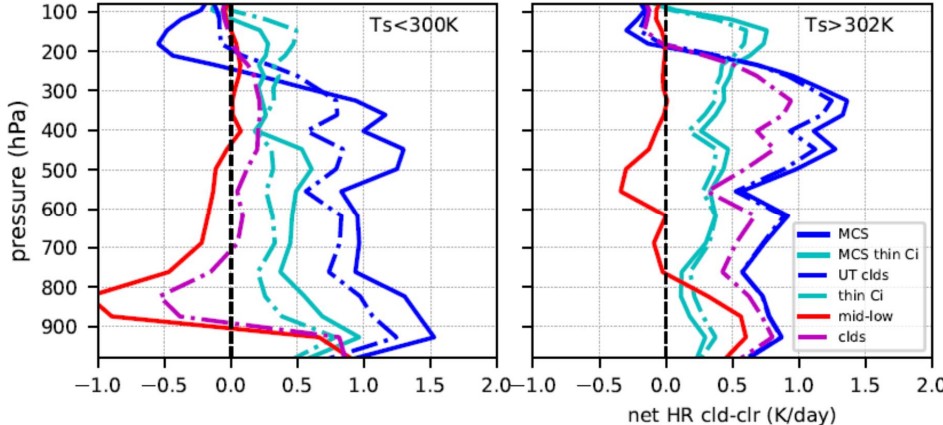

**Figure 9: Tropical 24-hour mean cloud net radiative heating effect (magenta) within the troposphere, and separately the effect of low- and mid-level clouds (red), mesoscale convective systems (blue), thin cirrus (cyan) and all UT cloud systems (black), averaged over 15N to 15S, when clouds or the specific cloud types are present. Left: regions with $T_{surf} < 300$ K, right: regions with $T_{surf} > 302$ K.**





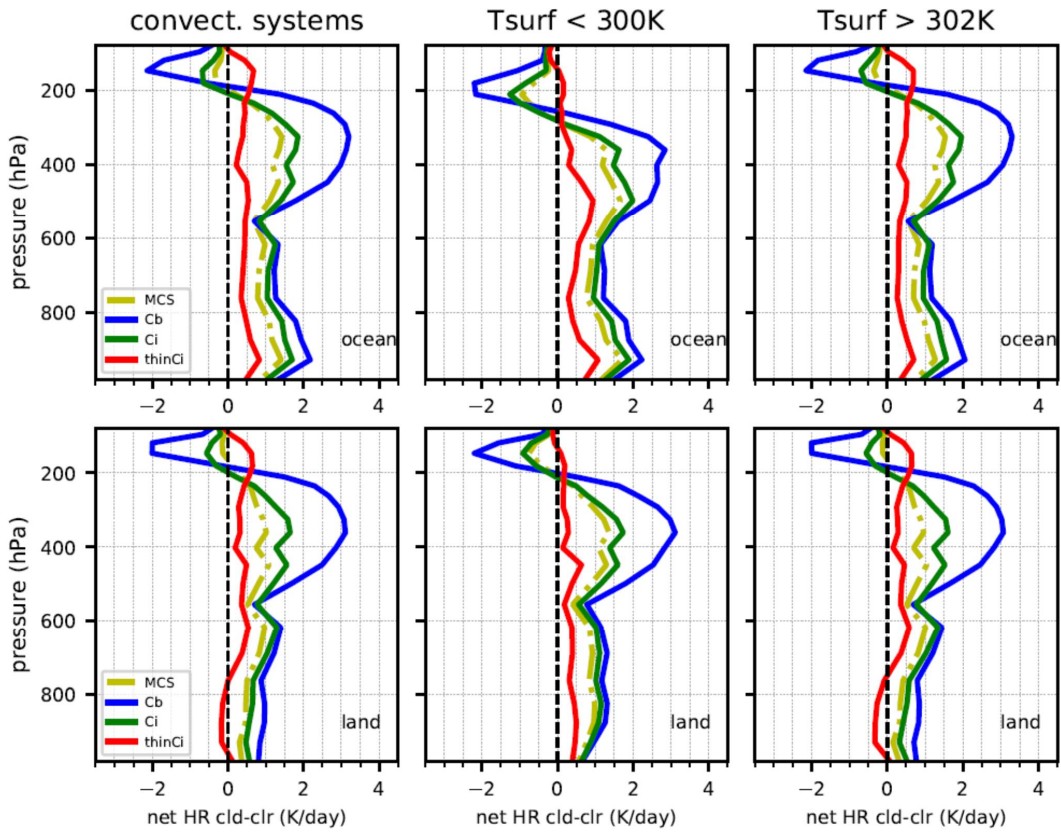

**Figure 10:** Mean net radiative heating effect of MCSs, when present, and their convective cores (Cb), cirrus anvil (Ci) and surrounding thin cirrus (thinCi) over ocean (top) and over land (with $p_{surf}$ > 900 hPa, bottom), at 1:30 AM local time. In addition, MCSs with the 25% warmest areas of $T_{surf}$ < 300K and of $T_{surf}$ > 302K are distinguished (from left to right). Cloud observations at 1:30PM local time, with SW radiation normalized to 24 hours. Statistics of 15 years (2004-2018), averaged over 15N to 15S.

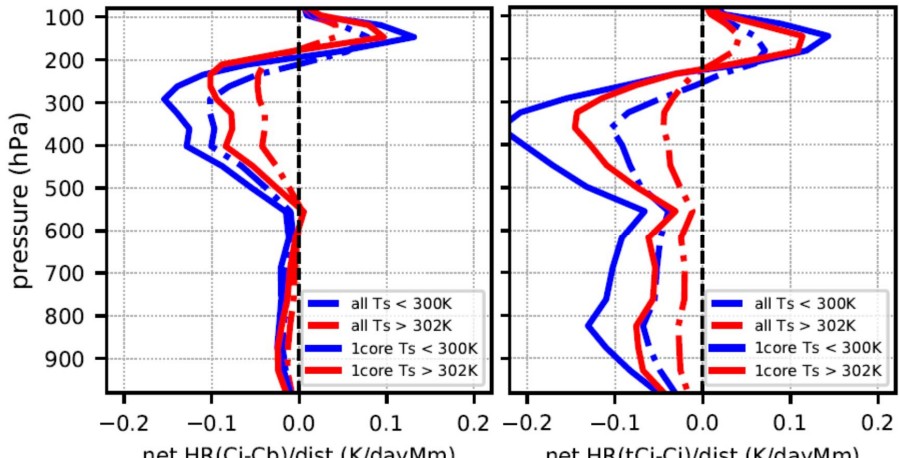

**Figure 11: Difference of 24-hr mean net radiative heating between cirrus anvil and convective cores (left) and between surrounding thin cirrus and cirrus anvil (right), divided by the distance between the centers of these MCS parts, assuming circular systems, for MCSs with the 25% warmest areas of $T_{surf} < 300K$ and of $T_{surf} > 302K$, respectively. Compared are all MCSs, defined as UT cloud systems with at least one convective core, and those with only one convective core with a coverage within the MCS of at least 5%. Cloud observations at 1:30PM local time, with SW radiation normalized to 24 hours. Statistics of 15 years (2004-2018), averaged over 15N to 15S.**





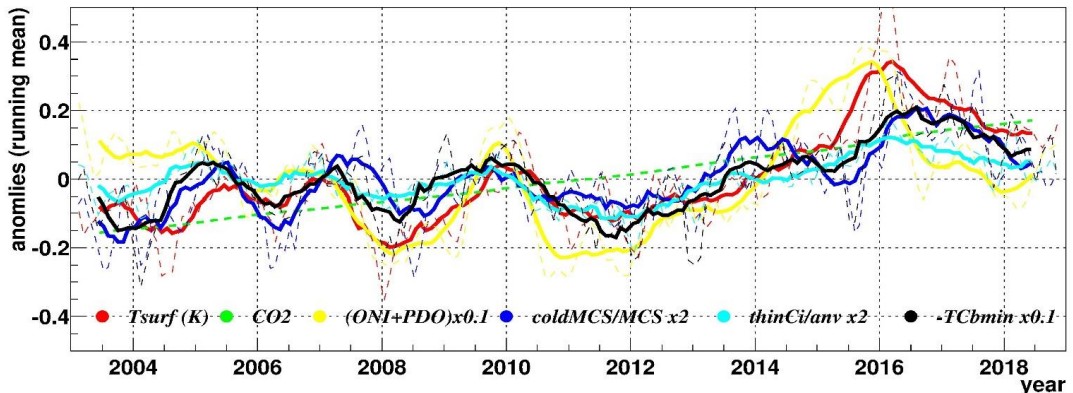

**Figure 12: Time series of 12-month running means (bold lines) and 3-month running means of anomalies of global surface**
**temperature (GISTEMP v4, Lenssen et al., 2019), ENSO index combined with Inter-decadal Pacific Oscillation index (each multiplied by 0.1), as well as coverage of cold MCSs over all MCSs (multiplied by 2), area of thin cirrus over area of total cirrus anvil (multiplied by 2) and minimum convective core temperature (in K multiplied by -0.1).**



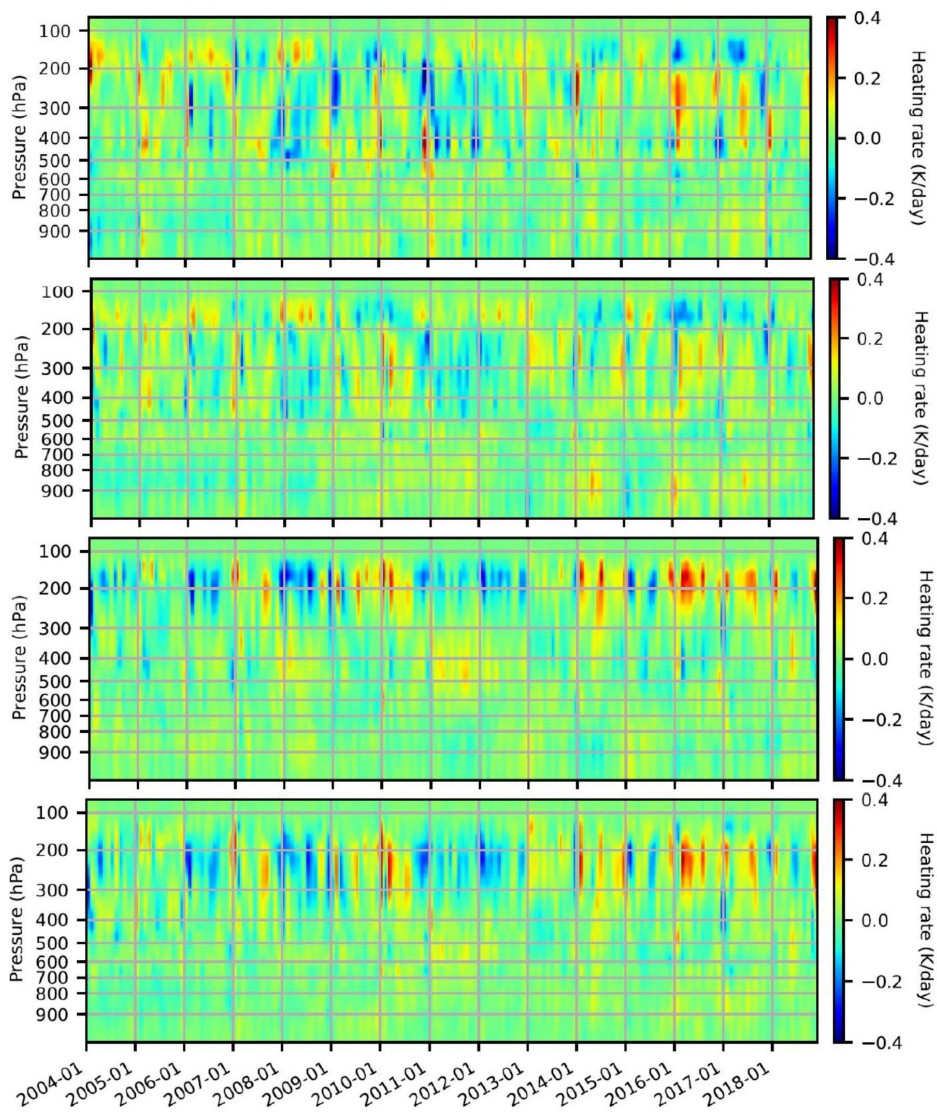

**Figure 13: Time series of deseasonalized anomalies of heating / cooling effect of MCSs, when present; from top to bottom: net during nighttime (1:30 AM LT), LW, SW and net during daytime (1:30 PM LT).**

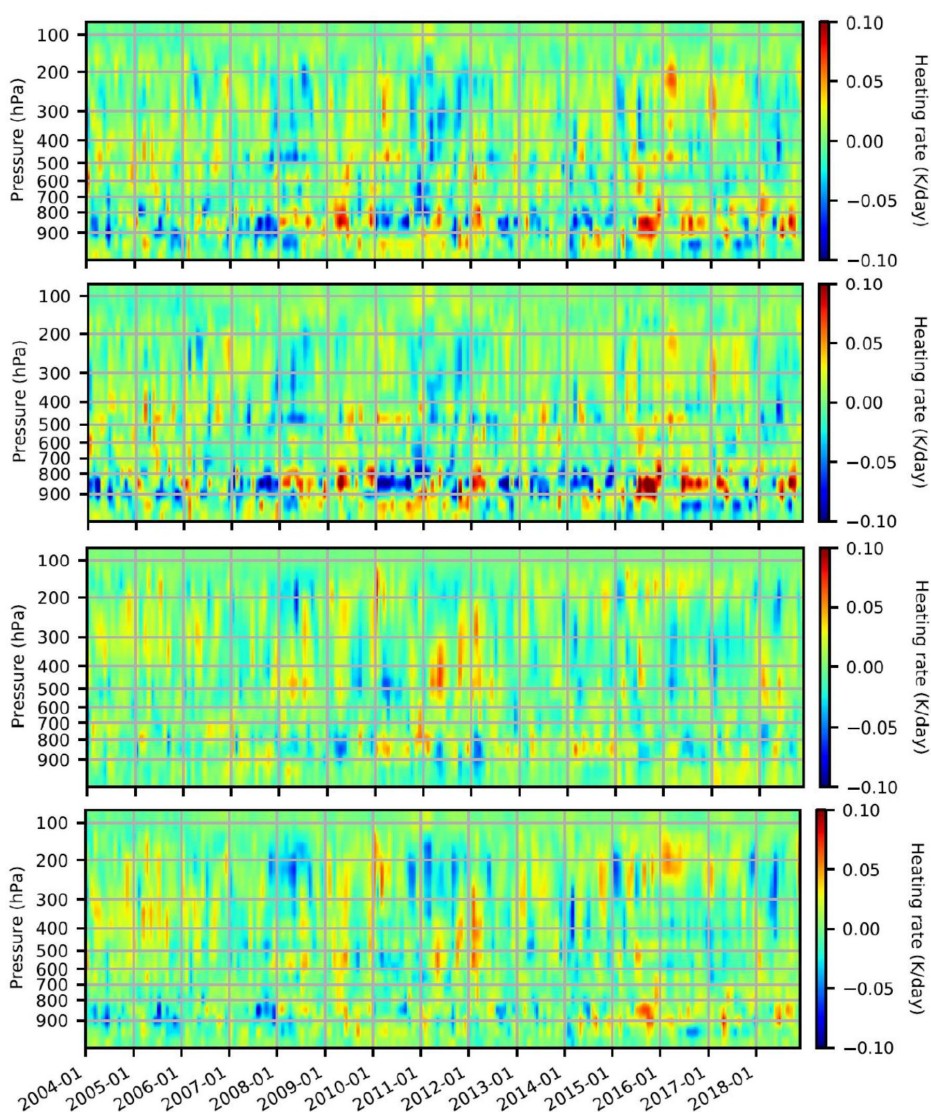

**Figure 14: Time series of deseasonalized anomalies of cloud heating / cooling effect; from top to bottom: net during nighttime (1:30 AM LT), LW, SW and net during daytime (1:30 PM LT).**
