# Peer review of "3D Radiative Heating of Tropical Upper Tropospheric Cloud Systems derived from Synergistic A-Train Observations and Machine Learning"

_Atmospheric Chemistry and Physics, 2020_

## Referee Comment (RC1) · Anonymous Referee #1 · 8 Sep 2020

3D Radiative Heating of Tropical Upper Tropospheric Cloud Systems derived from Synergistic A-Train Observations and Machine Learning

The authors present a method for proving 3D radiative heating structures using an ANN trained from the AIRS, CloudSat 2b-FLXHR-lidar product, CIRS cloud properties, and reanalysis environmental properties. This is a novel method for expanding the converge of actively-based products in order to describe the effect of upper tropospheric clouds on tropical radiative heating rates and their relation to surface temperature. The ANN method has been applied previously and for this case expands 3D heating rates

to a longer data record and increased spatial resolution to allow a more robust analysis on changes in upper tropospheric clouds and MCSs. The authors provide a thorough description of the retrieval and sources of error. It does seem crowded at times with the large amount of material and supplemental material. It would be good to streamline the use of extra material or to partition the addition retrieval/uncertainty aspects into a separate paper and move some supplemental material based on results to the main manuscript. Further, clarifications on some subject matters are needed and significance testing on the last two results sections is needed.

Major Comments

1. Looking at Fig 2 and Fig S4, there do seem to be some further physical explanations. For the LW it makes sense that error would be contained to cloud top in Cbs and Ci or just below cloud base in Ci. Below these regions the LW signal will likely be mostly impacted by the high RH in the tropical atmosphere. The SW signal does demonstrate variability below Ci cloud base ∼400 hPa. This could be errors in representation of Ci optical depth or clouds below the Ci reflecting SW back towards TOA. Multi-layer structures are essential to represent in Ci and thin-Ci in the tropics as the majority of cirrus contain a cloud below them (as in cited Hang et al 2019). Is there a way to capture if the ANN is representing the multi-layer structures below Ci? This is mentioned briefly around Line 445, but did not know if this was quantifiable.

2. Section 3.3. As mentioned in the text, during La Nina changes the location of cloud structures, but ENSO also significantly changes the size and occurrence of MCSs over the tropical oceans due to changes in the environment (e.g. Schumacher et al 2004; Henderson et al 2018; Stephens et al. 2018; Wodzkicki and Rapp 2020). During La Nina the MCSs are usually more isolated and less intense. This will likely have an impact on the observed cirrus cloud fractions. Is there a reason only one end of the ENSO spectrum was considered here? Does this case study limit the sampling of the structures?

Schumacher, C., R. A. Houze, and I. Kraucunas, 2004: The Tropical Dynamical Response to Latent Heating Estimates Derived from the TRMM Precipitation Radar. J. Atmos. Sci., 61, 1341–1358

Henderson, D. S., C. D. Kummerow and W. Berg, 2018: ENSO influence on TRMM tropical oceanic precipitation characteristics and rain rates. J. Climate, 31, 3979–3998

Stephens, G. L., and Coauthors, 2018: Regional intensification of the tropical hydrological cycle during ENSO. Geophys. Res. Lett., 45, 4361–4370

Wodzicki, K. R., and A. D. Rapp, 2020: Variations in Precipitating Convective Feature Populations with ITCZ Width in the Pacific Ocean. J. Climate, 33, 4391–4401

3. To aid the user, how much data needs to be averaged to obtain a representative heating profile? ANNs can give a statistically representative answer, but it might take some averaging to remove the random noise. How much data, spatial and temporal, need to be averaged to remove random error and get an accurate result?

4. Sec 4.2: Using warm regions and cool regions is a good way to initially separate these, but I would be careful with relating differences based on surface temperature. Other main factors, such as local environment and dynamical influences will also need to be considered. For example, MCSs in the West vs East Pacific are quite different in both surface temperature and structure due to thermal forcing in the West Pac and more dynamical forcing in the East Pac due to strong SST gradients. Further, as mentioned above (and in Section 4.3), ENSO can play a large role in the shape of MCSs due to changes in environment and regional dynamics (e.g. Schumacher et al 2004; Henderson et al 2018; Wodzkicki and Rapp 2020). Are the two surface temp (300K vs 302K) categories here consistent in the way MCSs would be initiated? Would isolating the same comparison to a similar region yield similar differences in characteristics?

5.

Section 4.3 I do not think this data record is long enough to make a significant regression analysis. It is OK that these results are here, but a stronger statement on how these results seem to be linked to changes in the ONI+PDO needs to be made and that it could change with a longer record. Do the regressions change if you break up the time periods (e.g. 2003-2012; 2007-2015; 2011-2018)? If significant regressions cannot be found the observed change with surface temperature is more likely due to natural variability. Adler et al (2017) stated that natural variability is too large to make statements on temperature and data periods longer than 30 years are needed.

Adler, R. F., G.Gu, M.Sapiano, J.-J.Wang, and G. J.Huffman, 2017: Global precipitation: Means, variations and trends during the satellite era (1979–2014). Surv. Geophys., 38, 679–699

6. The MCSs are defined using the presences of UT clouds and a convective core. How do you deal with cases were an MCS extends through multiple boxes? How do you ensure that cirrus is not associated with a nearby MCS and in proximity to isolated convection?

7. The usage of supplement material needs to be streamlined somehow. There is a lot of material overall and at some points this feels like two papers that have been pushed together: one outlining the retrieval and performance and another applying the data. There is a lot of back and forth between the manuscript and supplementary material and supplemental figures seem too incorporated into the material. An example of this is the comparison of Fig. 5 and Fig. S7 or the additional information in S12 and S13. The authors compare the shapes of the heating profiles and it requires bouncing back and forth between the Supplemental and normal figures. It is described in text, but it is more useful to see the visual comparisons. Some of the Supplemental figures need to be added to main text if referenced (e.g. S12 or S13). Perhaps discussion on the performance could be added to the supplemental pages and then have the readers sent to supplemental to learn more.

Minor Comments

Line 237: When describing the case sensitivities, it is hard to follow. A table might be easier to visualize.

Line 255: For future analysis, converting to something like sigma vertical coordinates may help mitigate this issue.

Table 2 with MAE: It is hard to understand the magnitude of the error here. What is mean heating compared to the error? Fig S4, gives a slight example, but examples in the text would be useful.

Line 375: "The small cooling around 550 hPa is due to melting" – evidence for this?

Section 4.2 I would remind the readers here that this data is much longer than other vertically resolved datasets.

Figures: Differentiating solid vs broken lines is difficult in the legends.

Line 544: Is T < 210 K cloud top temperature?

---

## Referee Comment (RC2) · Anonymous Referee #2 · 3 Nov 2020

**Review von ACP paper:**

*3D Radiative Heating of Tropical Upper Tropospheric Cloud Systems derived from Synergistic A-Train Observations and Machine Learning*

The authors present a new method using ANN to expand narrow track CloudSat and CALIPSO data of heating rates using CRIS (Clouds from IR Sounders) and reanalyses data in order to generate a 3D heating rate data set for the inner tropics (15N to 15S). For the training of the ANN 4 years data collocated in space and time are used. The trained ANN are applied to a long term data yielding to a 15 year data set of 3D heating rates. This data set was discussed in the light of ENSO for different could type and cloud systems (e.g. MCS: mesoscale convective systems).

This principal approach using measurements collocated with 2 or 3D satellite data is not new and was applied for the generation of different products with higher resolution in space and time and expansion to longer data records more suitable for thorough analysis. The data sets used are described for some parts in too much detail while for the ANN part some details are missing. Some streamlining of the text would be good to improve the understanding by merging with the supplement part or move some parts of the detailed data set descriptions to the supplement including more text and description to the figures. The discussion of the results e.g. with ENSO index and PDO is mainly qualitatively, some quantitative measures should be added.

**Major comments:**

1. Section 2.1: This section should be shortened and only the key facts of the CRIS data set relevant for the interpretation of the results should be mentioned. The other part can be moved to the supplement part. Line 90: Is the information about AMSU relevant for CIRS, if yes, this yields to some restrictions for the application of the new ANN-based method.

2. Section 2.2: This section can also be shortened or some parts can be moved to the supplement. E.g. the ERA-Interim description, TIGR data set.

3. Section 2.3: This section can also be shortened or some parts can be moved to the supplement.

4. Line 223 ff: the absolute number of pattern (samples for training/test/validation) should be given. These values are important for the interpretation of the results in Tab. 2 as well as for Fig. S1 and S2. For the latter ones, it should be explained why the number of epochs is different for the different data sets and what was the stopping criteria for the training of the ANN.

5. Line 280 ff.: The different types of models should be given in a bullet list or table with corresponding labels given in Tab. 2 and streamlined with the labels in Fig. S1, S2.

6. Tab. 2 & 3: In these tables as well as in the discussion of it, relative error measures should be given, too. It would be good in order to judge the approximation and generalisation accuracy of the ANN it would be good to have the mean absolute percentage error (MAPE) in addition.

7. Fig 13 & 14: It is hard to compare the different panels of these figures and the usage e.g. for upcoming climate studies. It would be better to have only one panel of the total net HR. Furthermore in order to judge the influence of the ENSO index, PDO and surface temperature (see Fig. 12) to total

net HR over time in a more quantitative way, mean total net HR time series data for different pressure layers (e.g. low, middle, high) should be correlated to the time series data of Fig. 12.

**Minor comments**

1. Line 255 ff. There are techniques available to deal with partly missing values in the target vector. The target vector can be masked for valid/not valid training value in the target vector. Then only for the valid elements in the target vector, the error in backpropagated during training. For not valid elements (NaN) the error is set to zero. This is a proven concept for training of ANN with incomplete target vectors.

1. Line 423 & 424: ".. is 24% larger, larger than 21% found by Li … " needs some clarification

2. Line 578: data processed for 30N to 30S; but only results of latitude band 15N to 15S are shown (Fig. 10).

3. Fig. 7: the data sources should be mentioned in more detail.

4. The quality of some figures should be improved e.g. Fig. 6 (use of vector instead of raster graphics is highly recommended)

**Recommendation:**

The developed method to derive high resolution 3D HR in the inner tropics uses a lot of different model data: CRIS, ERA-interim, MOIDS AOD. Each of the models mentioned has its own errors and bias which are described well in paper. ANN can handle systematic model error of the input data well, but if one or more models will change over time (which is likely for such kind of long term data sets) the trained ANN model for the generation of 3D HR data will generate most likely biases. ANN are also not able to cope for random errors in the model input data.

This can be omitted if the original satellite data (in this case AIRS spectral radiance data) are used with full spectral resolution as input data. This makes the ANN HR model more applicable and more robust especially in order to transfer this approach to other IR sounder data (e.g. IASI) for further studies. For transfer of a trained ANN model on AIRS data e.g. transfer learning techniques can be used to adapt it for IASI.

---

## Author Comment (AC1) · 19 Nov 2020

**Response to Reviewer #1**

We thank the Reviewer #1 for his/her positive and thorough assessment of the manuscript and for the thoughtful and constructive comments.

Indeed, in the beginning we considered writing a 2-part article, but several reasons made us finally decide to submit the results in a single article. First of all, ACP does not publish technical articles, so we would have needed to submit the two parts to different journals, which would have made the review process even more complicated, and secondly, the ANN method itself is a well-established method. The novelty consists in its application by training the ANNs over a large statistics of collocated data, though limited in space and time, to develop optimized non-linear regression models to provide a more complete picture in space and time. The provided comments helped us to make the whole manuscript easier to follow. Where appropriate, we modified the text of the manuscript and the supplement with the changes marked in yellow, at the end of the response,

With the point-by-point answers to each of the comments.

*Major comments*

*1. Looking at **Fig 2 and Fig S4**, there do seem to be some further physical explanations.*

*For the LW it makes sense that error would be contained to cloud top in Cbs and Ci or just below cloud base in Ci. Below these regions the LW signal will likely be mostly impacted by the high RH in the tropical atmosphere. The SW signal does demonstrate variability below Ci cloud base 400 hPa. This could be errors in representation of Ci optical depth or clouds below the Ci reflecting SW back towards TOA. Multi-layer structures are essential to represent in Ci and thin-Ci in the tropics as the majority of cirrus contain a cloud below them (as in cited Hang et al 2019). Is there a way to capture if the ANN is representing the multi-layer structures below Ci? This is mentioned briefly **around Line 445**, but did not know if this was quantifiable.*

Thank you for these extra explanations. We have improved the text accordingly in section 3.1. First of all, we have moved Fig. S3 to the main manuscript as Fig. 2. This makes it also easier to follow the discussion (see also response to major comment 7). Though we do not have any information on the cloud vertical extent, we could demonstrate that the cloud emissivity is closely related to the cloud vertical extent. It has also been shown that cloud vertical extent and number of vertical cloud layers are related (e.g. Wang et al. 2000). The neural networks seem to catch these dependencies quite well. Nevertheless, uncertainties due to these two variables which are not directly given in the input data are reflected afterwards in the predicted HRs.

Recently, we have developed ANN models to predict vertical extent and even a probability of a cloud-layer underneath, again using the same collocated AIRS-CALIPSO-CloudSat data (article in preparation). This allows a separate evaluation. For cloud vertical extent, the bias is 0 km and the standard derivations between predicted and observed vertical extent can be interpreted as uncertainties for Ci (thin Ci) of 38% (32%) over ocean and 43% (37%) over land. The hitrate for the probability of multiple cloud-layers is for Ci (thin Ci) 68% (66%) over ocean and 67% (68%) over land.

Another reason for the uncertainties is the variability of the vertical profiles of ice water content and ice crystal size distributions, which have been used to determine the FLXHR HRs, but again the input variables, like cloud emissivity and spectral cloud emissivity difference between 9 and 12 micron, only give indirect information on these.

Wang, J., W.B. Rossow, and Y. Zhang, 2000: Cloud Vertical Structure and its Variations from a 20-Yr Global Rawinsonde Dataset. J. Climate, 13, 3041-3056.

*2. Section 3.3. As mentioned in the text, during **La Nina changes the location of cloud structures**, but **ENSO also significantly changes the size and occurrence of MCSs over the tropical oceans due to changes in the environment** (e.g. Schumacher et al 2004; Henderson et al 2018; Stephens et al. 2018; Wodzkicki and Rapp 2020). **During La Nina the MCSs are usually more isolated and less intense**. This will likely have an impact on the observed cirrus cloud fractions. Is there a reason only one end of the ENSO spectrum was considered here? Does this case study limit the sampling of the structures?*

Schumacher, C., R. A. Houze, and I. Kraucunas, 2004: The Tropical Dynamical Response to Latent Heating Estimates Derived from the TRMM Precipitation Radar. J. Atmos. Sci., 61, 1341–1358
Henderson, D. S., C. D. Kummerow and W. Berg, 2018: ENSO influence on TRMM tropical oceanic precipitation characteristics and rain rates. J. Climate, 31, 3979–3998
Stephens, G. L., and Coauthors, 2018: Regional intensification of the tropical hydrological cycle during ENSO. Geophys. Res. Lett., 45, 4361–4370
Wodzicki, K. R., and A. D. Rapp, 2020: Variations in Precipitating Convective Feature Populations with ITCZ Width in the Pacific Ocean. J. Climate, 33, 4391–4401

Section 3.3 still belongs to the technical part. We wanted to show how for one month of data the predictions over the full CIRS swath compare to the FLXHR data over the nadir tracks. The geographical maps in Fig. 4 were only meant as an illustration, but indeed by emphasizing in the beginning that the chosen month corresponds to a La Nina situation one could have expected a comparison between El Nino and La Nina. We have changed this in the text. As you point out, there have been already so many publications about ENSO, that a new study should be more profound and is therefore beyond the scope of this publication (we added a phrase in section 4.3 with references).

*3. To aid the user, how much data needs to be averaged to obtain a representative heating profile? ANNs can give a statistically representative answer, but it might take some averaging to remove the random noise. How much data, spatial and temporal, need to be averaged to remove random error and get an accurate result?*

Indeed, the development of ANN regression models leads to reliable mean values. As we are interested to relate radiative heating rates of different cloud types, we have developed models separately per cloud type and separately per land and ocean to minimize biases between different scenes. As we see in the new Fig. 2 (old S3), the differences between predicted and observed radiative heating rate profiles are for all cloud types close to 0 K/day, whereas the 30% and 70% quantiles of the distributions indicate the uncertainty of individual predictions. The results in Figure 3, which presents monthly mean HR profiles for the different scenes averaged over the tropics, compare very well with the averages from the nadir

tracks. So this leads to the conclusion that monthly means over the tropics (or deep tropics) are well represented, even if one distinguishes cloud types or environmental conditions. Section 4 presents results averaged over 10 to 15 years or are shown as monthly means over the time series.

*4. Sec 4.2: Using warm regions and cool regions is a good way to initially separate these, but I would be careful with relating differences based on surface temperature. Other main factors, such as local environment and dynamical influences will also need to be considered. For example, MCSs in the West vs East Pacific are quite different in both surface temperature and structure due to thermal forcing in the West Pac and more dynamical forcing in the East Pac due to strong SST gradients. Further, as mentioned above (and in Section 4.3), ENSO can play a large role in the shape of MCSs due to changes in environment and regional dynamics (e.g. Schumacher et al 2004;*
*Henderson et al 2018; Wodzkicki and Rapp 2020). Are the two surface temp (300K vs 302K) categories here consistent in the way MCSs would be initiated? Would isolating the same comparison to a similar region yield similar differences in characteristics?*

We agree completely that surface temperature is only one variable important for the onset of convection. However, recently the average of the 30% warmest SSTs have been successfully used as a proxy for tropical convective activity by Fueglistaler (2019). Over ocean our thresholds of 300K and 302K correspond to the 30% coolest and warmest ocean regions, respectively, and indeed when comparing the SSTs underneath the opaque parts (emissivity > 0.9) of the MCSs and of the coldest MCSs in the new Fig. 7, we see that these SSTs are shifted 30% towards the warmest SSTs. We have added this figure and explanations in section 4.2.

Since for land with a larger diurnal cycle and more heterogeneity, the surface temperature alone does not give information on the convective activity, we have taken out the analysis over land.

By distinguishing cooler and warmer regions we wanted to show that the results based on the predicted HRs are as expected, which gives further confidence in this new dataset.

We have also explored specific regions, as suggested. The results are displayed in Figure S11 in the supplement, with a short discussion in section 4.2. The differences in the 24-hr net radiative heating effect profiles are larger between cool and warm regions within these regions than between the average regional profiles. From this one may conclude that the on average slightly increasing HR effects of the MCS from tropical Atlantic to West Pacific can mostly be explained by increasing parts of warm SSTs from the tropical Atlantic towards the West Pacific. Differences in dynamics and atmospheric environment certainly also play a role, but this is more on the process level.

Fueglistaler, S.: Observational Evidence for Two Modes of Coupling Between Sea Surface Temperatures, Tropospheric Temperature Profile, and Shortwave Cloud Radiative Effect in the Tropics, Geophys. Res. Lett., 46, 9890-9898, https://doi.org/10.1029/2019GL083990, 2019.

*5. Section 4.3 I do not think this data record is long enough to make a significant regression analysis. It is OK that these results are here, but a stronger statement on how these results seem to be linked to changes in the ONI+PDO needs to be made and that it could change with a longer record. Do the*

*regressions change if you break up the time periods (e.g. 2003-2012; 2007-2015; 2011-2018)? If significant regressions cannot be found the observed change with surface temperature is more likely due to natural variability. Adler et al (2017) stated that natural variability is too large to make statements on temperature and data periods longer than 30 years are needed.*

*Adler, R. F., G.Gu, M.Sapiano, J.-J.Wang, and G. J.Huffman, 2017: Global precipitation: Means, variations and trends during the satellite era (1979–2014). Surv. Geophys., 38, 679–699*

We agree that the record is still too short to give conclusions on climate change. We have given changes with tropical surface warming, assuming linear relationships, for comparison to other publications which are based on data records of similar length (or even shorter). We have revised section 4.3 by using now the tropical surface temperature (which is strongly correlated with global surface temperature) and have also computed the Pearson correlation coefficients for a further indication of uncertainty. We have also compared the changes with tropical surface warming between 2003-2018 with those between 2003-2014 and 2007-2018. The tendencies are the same. Furthermore we have added a correlation analysis for radiative heating averaged in the upper, middle and lower atmosphere.

*6. The MCSs are defined using the presences of UT clouds and a convective core. How do you deal with cases where an MCS extends through multiple boxes? How do you ensure that cirrus is not associated with a nearby MCS and in proximity to isolated convection?*

The cloud system approach consists in merging adjacent grid boxes with high-level clouds (at least 65% grid coverage) of similar height ($\Delta p_{cld} < 50$ hPa) and is explained in an earlier article (Protopapadaki et al., 2017). We have collocated the cloud system data with the ones of the HRs, so that we can associate the HRs to all grid boxes of the UT cloud systems. For the analysis of MCS we ask for at least one convective core. Indeed, multi-core convective systems may be single core systems which are connected via thin cirrus, and indeed they may be separate systems in different phases of maturity. We provide comparisons between single core results and multi-core results in Fig. 10, and see that these differences are not large.

*7. The usage of supplement material needs to be streamlined somehow. There is a lot of material overall and at some points this feels like two papers that have been pushed together: one outlining the retrieval and performance and another applying the data. There is a lot of back and forth between the manuscript and supplementary material and supplemental figures seem too incorporated into the material. An example of this is the comparison of Fig. 5 and Fig. S7 or the additional information in S12 and S13. The authors compare the shapes of the heating profiles and it requires bouncing back and forth between the Supplemental and normal figures. It is described in text, but it is more useful to see the visual comparisons. Some of the Supplemental figures need to be added to main text if referenced (e.g. S12 or S13). Perhaps discussion on the performance could be added to the supplemental pages and then have the readers sent to supplemental to learn more.*

Thank you for your last remark! We have revised the main manuscript, by including the figures which are necessary for the main discussions in the text, and we have added the discussions corresponding to the supplementary figures in the supplement. We have also moved the discussion on the sensitivity studies

to the selection of scenes used for the training as a separate chapter in the supplement. In this way, the technical part (section 3) should be much easier to read.

In section 4, we have revised the figures to be more concise (averaged histograms over AM and PM), and showing $T_{surf}$-dependent results only over ocean (see point 4). This then allows us to add maps of HRs over 3 different atmospheric layers (< 200 hPa, 200 – 600 hPa, 600 – 900 hPa).

**Minor comments**

*Line 237: When describing the case sensitivities, it is hard to follow. A table might be easier to visualize.*
We moved the description of the sensitivity study over the scenes to the supplement and have emphasized the 4 cases by building a list within the text.

*Line 255: For future analysis, converting to something like sigma vertical coordinates may help mitigate this issue.*
This is a very interesting suggestion; thank you! One needs an investigation if the changing pressure coordinates allow a reliable training of the ANN over all land.

*Table 2 with MAE: It is hard to understand the magnitude of the error here. What is mean heating compared to the error? Fig S4, gives a slight example, but examples in the text would be useful.*
The problem with the MAE is that it is a one-dimensional variable. It is used in machine learning as a metrics to get the best fit. It is the mean of the absolute errors over the 22 p-layers. For the studies in Section 4 it is more important to know the uncertainties for each of the pressure levels and for each scene type. This is shown in Fig. 2, and the HR profiles constructed over the whole tropical band in comparison with the FLXHR nadir track statistics are shown in Fig. 3.

*Line 375: "The small cooling around 550 hPa is due to melting" – evidence for this?*
We found one reference about this (Johansson et al. 2015) which we cite in section 4; and in section 3 we changed to 'The small cooling around 550 hPa is due to melting, owing to the transition from liquid to ice phase which occurs at or just below the freezing level at about 5 km altitude throughout the tropics, and the different emissivities of liquid and ice cause a flux divergence at that level (Tristan L'Ecuyer, personal communication).'
I discussed this with Tristan L'Ecuyer in 2017, who wrote: 'This is in fact a radiative effect owing to the transition from ice to liquid phase in mostly the surrounding stratiform areas of convective systems. In that case most melting occurs at or just below the freezing level and the different emissivities of liquid and ice cause a flux divergence at that level, which happens to occur at 5 km throughout the year in the tropics.'

*Section 4.2 I would remind the readers here that this data is much longer than other vertically resolved datasets.*
Actually this is most important for section 4.3, when we show the time series, but you are right; our statistics is much higher than what was published before.

*Figures: Differentiating solid vs broken lines is difficult in the legends.*
Figures redone

*Line 544: Is T < 210 K cloud top temperature?*
The IR sounder retrieval does not provide the cloud top temperature, but corresponds to a height where the cloud reaches an optical depth of about 0.5, which corresponds to about $1.5 - 2$ km below cloud top (Stubenrauch et al. 2017). We have changed this in the text to 'cold MCSs with near-cloud top temperature $< 210$ K'.

[revised manuscript text omitted]
). The number of epochs to converge towards a minimum loss is relatively small: less than 60 for cloudy scenes (Figure S1) and less than 45 for clear sky scenes (Figure S2). Essentially, the MAE decreases considerably only within the first 10 (5) epochs for cloudy (clear sky) scenes. The relatively small number of epochs necessary for convergence may be explained by the large statistics we use for the training and the number of relevant variables for the prediction. The final choice of parameters corresponds to Npar40 / Npar41 in Figure S1, as the MAE is smallest and comparable with the ones of Npar41 / Npar42 and Npar42 / Npar43. For clear sky scenes, Figure S3 compares the evolution of MAE for models developed over ocean over land and over both. Figure S2 shows that predictions over ocean will be better than over land which can be explained by a better homogeneity. Figure S2 also shows that by using the atmospheric profiles with a better vertical resolution (20 layers instead of 10) does not improve the results.

930

935

940    **For LW / SW:      Npar18 / Npar19 = basic, Npar20 / Npar21 =+CR, Npar30 / Npar31 =+RH10**
**Npar40 / Npar41 =+T10,  Npar41 / Npar42 =+w500,    Npar42 / Npar43 =+AOD**

[Figure]

**Figure S1: Sensitivity results concerning surface, atmospheric and cloud input parameters for the prediction of LW (left) and SW (right) radiative heating rates of clouds over ocean: Mean absolute error (in K/day) of training (dots) and validation (lines) for the experiments 1-6 for LW and 2 – 6 for SW, using the parameters listed in Table 1.**

945

**For LW / SW:**  ocean_Npar35 / Npar35 = clear sky basic without cloud properties + clear sky fraction of CR + RH10 + T10,
land_Npar38 / Npar36 = as for ocean, + 3 IR surface emissivities / + 1 surface albedo
Npar39 / Npar37 = for land and ocean together, including land-ocean flag
Npar60 / Npar58 = for land and ocean together, RH20 + T20 instead of +RH10+T10

[Figure]

950

**Figure S2: Sensitivity results for the prediction of LW (left) and SW (right) radiative heating rates of clear sky scenes as determined by CIRS: Mean absolute error (in K/day) of training (dots) and validation (lines): for the parameters listed in Table 1, excluding cloud properties and their uncertainties.**

955

**Sensitivity to the selection of scenes used for the training**

These sensitivity studies are dedicated to the scene types for which we develop the models:

    i)       all clouds over the whole tropical band (one model)

    ii)     all clouds separately over ocean and over land (two models)

    iii)    high-level clouds and mid- / low-level clouds individually, each separately over ocean and over land (four models)

    iv)    Cb, Ci / thin Ci, mid- / low-level clouds individually, each separately over ocean and over land (six models)

In addition, we develop models for clear sky i) over the whole tropical band (one model) and ii) separately over ocean and over land (two models). In general, a model trained over all scenes together soothes out differences between different cloud types and between ocean and land. Also scenes which are less frequent may have a smaller weight and may be therefore less represented than other scenes. Since we are interested in the study of the effect of UT cloud systems, we choose to use separate models. In particular, the modelling of Cb clouds is improved when exploiting a dedicated training for this cloud type, which represents about 7% of all clouds in the deep tropics (Stubenrauch et al., 2017).

To illustrate the effect of model aptness in dependence of training scenes, we compare in Figure S3 the difference between the predicted radiative HRs and those from CALIPSO-CloudSat over ocean for Cb, cirrus and thin cirrus, in the LW and SW, respectively. Compared are models which were trained i) for all clouds over ocean and land together, ii) for all clouds over ocean, iii) for high-level clouds over ocean and iv) for Cb and for Ci / thin Ci over ocean. All results are quite similar, with the differences between mean predicted and 'observed' radiative HRs undulating well around 0 K/day. However, we observe an overestimation of the LW cooling above Cb clouds by nearly 1 K/day when all clouds together are used to develop one single model. The results improve for cirrus and thin cirrus when a dedicated model is developed for these cloud types. For the SW HRs it is not possible to determine the best performance among these four models. The SW heating in the upper part of Cb clouds is more difficult to predict, as for all four models the mean difference undulates around 0 K/day within ± 0.8 K/day between 100 and 200 hPa. Considering the radiative HR profiles shown in Figure S4 of the supplement, we find that the largest uncertainties for Cb clouds are around the maxima of LW cooling and SW heating. Furthermore, we observe that all models give very similar results, so that in the following we will mainly use the most specific scene models, leading to the application of eight models to reconstruct the radiative heating rate fields over the tropics.

We have also estimated the uncertainty related to the choice of scenes for the training after having applied these different ANN models to one month of AIRS data, over the whole tropical band (30N – 30S). Figure S5 presents for three atmospheric layers the difference between predicted LW HRs obtained from four models (clouds over ocean, clouds over land, clear sky over ocean, clear sky over land) and from two models (clouds and clear sky) and between predicted LW HRs obtained from all final eight models and from the four scene-dependent models. These differences, which give an indication of the uncertainty, lie

generally within 0.25 K/day, with only a few regions of 0.45 K/day, keeping in mind that the most detailed scene distinction will give the better results.

990

[Figure]

**Figure S3: Sensitivity results concerning training over different scenes (high clouds over ocean, all clouds over ocean and all clouds over ocean and land) for the prediction of high-level cloud LW radiative heating rates (above) and SW radiative heating rates (below): difference between predicted and observed vertical profiles, separately for Cb, Cirrus and thin Cirrus, as identified by AIRS-CIRS, over tropical ocean. 30% and 70% quantiles of the distributions are also shown.**

[Figure]

Figure S4: Sensitivity results concerning training over different scenes (high clouds over ocean, all clouds over ocean and all clouds over ocean and land) for the prediction of high-level cloud LW radiative heating rates (above) and SW radiative heating rates (below): predicted vertical profiles compared to those from CALIPSO-CloudSat (black lines), separately for Cb, Cirrus and thin Cirrus, as identified by AIRS-CIRS, over tropical ocean. 30% and 70% quantiles of the distributions are also shown.

[Figure]

**Figure S5: LW heating rate differences in 3 layers (106-131 hPa, 200-223 hPa, 525-585 hPa) between combination of (left) 4 models (clouds over ocean, clouds over land, clear sky over ocean, clear sky over land) and of 2 models (clouds and clear sky) and (right) 8 models and 4 models (clouds over ocean, clouds over land, clear sky over ocean, clear sky over land).**

1015

Since the AIRS clear sky identification may also include subvisible cirrus as well as partly cloudy scenes within the AIRS footprint, we estimated how much this affects the radiative HRs by comparing the FLXHR-lidar HRs for AIRS clear sky and for CALIPSO-CloudSat clear sky identification (no GEOPROF-lidar cloud layer within the footprint). Definitely, Figure S6 shows a slight positive bias in the clear sky LW heating near 100 hPa of about 0.1 to 0.2 K/day due to subvisible cirrus, in particular during night, when the CALIPSO lidar better detects subvisible cirrus. The SW clear sky heating positive bias of the same order of magnitude between 200 and 800 hPa and the cold LW clear sky heating negative bias around 900 hPa are most probably linked to contamination by partial cloudiness. As our present HR data are stored at a spatial resolution of 0.5°, we have identified another bias, which is due to the identification of clear sky at a spatial resolution of 0.5°. Clear sky HRs are sampled only over grid boxes with all AIRS footprints identified as clear sky. The broken lines in Figure S6 present the difference between the average HRs, deduced by machine learning and averaged over 0.5° for cases with clear sky fraction equal to 1, and the average FLXHR-lidar HRs along the nadir tracks for the cases with CALIPSO-CloudSat clear sky identification. This gives an estimation of the biases due to sampling at coarse spatial resolution and effects on the machine learning.

1020

1025

[Figure]

1030

**Figure S6: Bias in LW and SW heating rates due to uncertainties in CIRS clear sky identification (full line), given as mean difference of FLXHR radiative heating rates of clear sky identified as no CloudSat-lidar GEOPROF cloud layers and of 'clear sky – partly cloudy' identified by AIRS using the CIRS 'a posteriori' cloud detection, and in addition the effect of sampling for clear sky fraction 1 over 0.5 ° (broken line), separately at nighttime (1:30 AM LT) and at daytime (1:30 PM LT). Statistics is over January 2008.**

1035

[Figure]

**Figure S7: Tropical mean latent heating (black), digitized from Figure 9 of Li et al. (2013), and tropical mean diabatic heating (red) as the sum of latent heating and net radiative heating (from Figure 5), including uncertainties due to cloud cover variation (dotted), LW HR variability between night and day (broken) and clear sky identification bias (only visible as slightly larger contribution near the surface and near the troposphere).**

1040

[Figure]

**Figure S8: Tropical mean cloud net radiative heating effect (magenta) within the troposphere above ocean at 1:30 AM LT (top) and at 1:30 PM LT (bottom), as well as the separate effects of low- and mid-level clouds (red), all UT clouds (blue dash-dotted), thin cirrus (cyan dash dotted), MCS (blue full line) and thin cirrus associated with MCS (cyan full line), averaged over 15N to 15S, when the specific cloud types are present. Left: regions with SST < 300 K, right: regions with SST > 302 K.**

[Figure]

Figure S9: Mean net radiative heating effect of maritime MCSs, when present, and their convective cores (Cb), cirrus anvil (Ci) and surrounding thin cirrus (thinCi) at 1:30AM LT (top) and at 1:30 PM LT (bottom). Compared to MCSs over cool areas (SST < 300K) and to MCSs over warm areas (SST > 302K). Statistics of 15 years (2004-2018).

[Figure]

1060

**Figure S10: Normalized distributions of surface temperature (ERA-Interim) over tropical ocean (20N-20S) and of surface temperature underneath the opaque part (cloud emissivity > 0.9) of all MCSs and of cold MCS (cloud temperature of opaque part < 210K). The black lines correspond to the threshold temperatures for the 30% coolest and 30% warmest surface temperatures, 300K and 302K, respectively. 15 years statistics averaged over 1:30 AM and**
1065 **1:30 PM LT.**

[Figure]

**Figure S11: Mean 24-hr net radiative heating effect of maritime MCSs, when present, and their convective cores (Cb), cirrus anvil (Ci) and surrounding thin cirrus (thinCi). First panel from left to right: over four specific regions (West Pacific: 12N-12S and 130E-170E; Central Pacific: 10N-10S and 150W-180W; Eastern Pacific: 10N-10S and 100W-130W; Atlantic: 10N-10S and 5W-35W). Second and third panel: each of these regions separately over cool areas (SST < 300K) and over warm areas (SST > 302K). Statistics of 15 years.**

---

## Author Comment (AC2) · 19 Nov 2020

**Response to Reviewer #2**

We thank the Reviewer #2 for the thoughtful and constructive comments concerning our manuscript.

Indeed, the ANN method itself is a well-established method. The novelty consists in its application by training the ANNs over a large statistics of collocated data, though limited in space and time, to develop optimized non-linear regression models to provide a more complete picture in space and time. The provided comments helped us to improve the manuscript for clarity. Where appropriate, we modified the text of the manuscript and the supplement with the changes marked in yellow. This marked text together with the new figures are provided at the end of the response to reviewer #1. Below, we provide point-by-point answers to each of the comments of reviewer #2.

*Major comments*
*1. Section 2.1: This section should be shortened and only the key facts of the CRIS data set relevant for the interpretation of the results should be mentioned. The other part can be moved to the supplement part. Line 90: Is the information about AMSU relevant for CIRS, if yes, this yields to some restrictions for the application of the new ANN-based method.*

First, we have added a short paragraph just after the title of section 2, which gives the purpose of the subsections. Section 2.1 describes the CIRS cloud data which are used as input for the machine learning as well as the cloud system data derived from the CIRS data which are used in the analysis of section 4. As both datasets are already published, we have shortened the whole section.

No, we don't use AMSU data. The sentence was there only to explain the grouping of 3 x 3 AIRS measurements. As this information is not relevant for the rest of the article, we have taken out this sentence.

*2. Section 2.2: This section can also be shortened or some parts can be moved to the supplement. E.g. the ERA-Interim description, TIGR data set.*

We have considerably shortened this section, as the ERA-Interim data are published elsewhere. We describe shortly all variables which are used as input parameters in the ANN models. However, we did not move the removed parts to the supplement, as the supplement is already quite long.

*3. Section 2.3: This section can also be shortened or some parts can be moved to the supplement.*

This section describes the target data as well as their quality. We also shortened this section, but we kept the description of how the radiative heating rates were determined as well as a summary of their evaluation.

All in all, we restructured and shortened sections 2.1-2.3 by more than 20% and hope that the new description is easier to read.

*4. Line 223 ff: the absolute number of pattern (samples for training/test/validation) should be given. These values are important for the interpretation of the results in Tab. 2 as well as for Fig. S1 and S2. For the latter ones, it should be explained why the number of epochs is different for the different data sets and what was the stopping criteria for the training of the ANN.*

The four years of collocated data correspond to a very large statistics of more than 16 million data points. We added this information in the first paragraph of section 2.4. When separating by scene type, the samples vary from 94000 Cb samples over land to 4.8 million mid- and lowlevel cloud samples over ocean. These samples contain both, AM and PM data. For the training of the SW heating rates, only half of the data are used (PM), which still leaves very large samples.

The number of epochs to converge towards a minimum loss is relatively small: less than 60 for cloudy scenes (Figure S1) and less than 45 for clear sky scenes (Figure S2). Essentially, the MAE

decreases considerably only within the first 10 (5) epochs for cloudy (clear sky) scenes. The relatively small number of epochs necessary for convergence may be explained by the large statistics we use for the training and the number of relevant variables for the prediction.

*5. Line 280 ff.: The different types of models should be given in a bullet list or table with corresponding labels given in Tab. 2 and streamlined with the labels in Fig. S1, S2.*
We have summarized the sensitivity experiments with the corresponding variables also in Table 1. In the supplement we added a text for the description and interpretation for Figures S1 and S2, as well as an assignment of the labels to the experiments.

*6. Tab. 2 & 3: In these tables as well as in the discussion of it, relative error measures should be given, too. It would be good in order to judge the approximation and generalisation accuracy of the ANN it would be good to have the mean absolute percentage error (MAPE) in addition.*
As we discuss in section 3, the average MAE over the vertical HR profiles is only one criterion to choose the best model for the prediction of the vertical HR profiles. In the beginning we also considered percentage errors, but the problem is that all cloud types and clear sky have at some vertical layer a value near 0. In particular for Cb, the lower layers have HRs close to 0. This would make an interpretation of the MAPE quite difficult. MAPE values would be automatically larger for profile types with more small HR values within the troposphere. In order to get reasonable MAPE values one has to introduce a lower absolute limit of the HR values. Considering the new Figures S3 and S4, which present the difference between predicted and observed HR profiles as well as the HR profiles, for Cb for example the difference is close to 0 for layers below an altitude of 800 hPa, while their HRs are also close to 0, leading to an artificially large percentage error. For Cb the maximum LW HR bias of the Cb model (red) is about 0.25 K/day for an average LW HR of -3.5 K/day and the maximum SW HR bias is about 0.5 K/day for an average SW HR of about 4 K/day. This corresponds to a percentage error of 7% and of 13%, respectively. As we use the same metrics in the tables, we can compare the performances of the models using different sets of variables. Figures S3 and S4 make it possible to roughly compute the percentage errors for different layers. MAPE may be a very useful metrics for other applications, but we do not see what the additional computation of MAPE would add to the interpretation of our results.

*7. Fig 13 & 14: It is hard to compare the different panels of these figures and the usage e.g. for upcoming climate studies. It would be better to have only one panel of the total net HR. Furthermore in order to judge the influence of the ENSO index, PDO and surface temperature (see Fig. 12) to total net HR over time in a more quantitative way, mean total net HR time series data for different pressure layers (e.g. low, middle, high) should be correlated to the time series data of Fig. 12.*
We now only show the 24-hr net radiative heating effects. In addition, we have added time series of the net HRs integrated over three vertical layers (100 - 200 hPa, 200 – 600 hPa, 600 – 900 hPa) and have computed correlation coefficients with the different other variables.

**Minor comments**
*Line 237: Line 255 ff. There are techniques available to deal with partly missing values in the target vector. The target vector can be masked for valid/not valid training value in the target vector. Then only for the valid elements in the target vector, the error in backpropagated during training. For not valid elements (NaN) the error is set to zero. This is a proven concept for training of ANN with incomplete target vectors.*
Thank you for this information. The authors were not aware of this, even after having discussed with several AI experts. Therefore we have used another method (replacing invalid values below

the surface by mean values classified per month and scene type), which is perhaps less elegant but should give similar results.

*Line 423 & 424: ".. is 24% larger, larger than 21% found by Li ... " needs some clarification*
We have redone the computations by using a LW average over AM and PM (instead of only PM) and added uncertainty estimates from clear sky identification and diurnal cloud amount variability. The final result lies between 20 and 25%, or 22 ± 3 %, which is only slightly larger than 21%. However the shape of the HR profile is different compared to the result of Li et al..

*Line 578: data processed for 30N to 30S; but only results of latitude band 15N to 15S are shown (Fig. 10).*
We have added maps which show results between 30N and 30S. We have mostly shown results for the deep tropics, as these have been shown by Li et al. 2013.

*Fig. 7: the data sources should be mentioned in more detail.*
Precipitable water and surface temperature are from ERA-Interim and UT cloud frequency of occurrence from CIRS.

*The quality of some figures should be improved e.g. Fig. 6 (use of vector instead of raster graphics is highly recommended)*
redone.

***Recommendation:***
*The developed method to derive high resolution 3D HR in the inner tropics uses a lot of different model data: CRIS, ERA-interim, MOIDS AOD. Each of the models mentioned has its own errors and bias which are described well in paper. ANN can handle systematic model error of the input data well, but if one or more models will change over time (which is likely for such kind of long term data sets) the trained ANN model for the generation of 3D HR data will generate most likely biases. ANN are also not able to cope for random errors in the model input data.*
*This can be omitted if the original satellite data (in this case AIRS spectral radiance data) are used with full spectral resolution as input data. This makes the ANN HR model more applicable and more robust especially in order to transfer this approach to other IR sounder data (e.g. IASI) for further studies. For transfer of a trained ANN model on AIRS data e.g. transfer learning techniques can be used to adapt it for IASI.*
We decided to use physical variables instead of radiances for different reasons. We use CIRS cloud data, retrieved from AIRS or IASI, together with ERA-Interim atmospheric and surface data. The latter have been also used as ancillary data in the CIRS retrieval, which gives a certain coherence. We tested two additional variables, vertical velocity from ERA5 and monthly mean AOD from MODIS, but finally we do not use them in the final models, as we could not detect a considerable improvement. We have been careful to only select variables which are also available for the CIRS-IASI data, so that the same models can be applied on IASI data. For the evaluation we need then independent data, as IASI and CALIPSO-CloudSat data do not overlap in the tropical band. We foresee to use ARM data for an independent evaluation, though these also have their issues, as mentioned in sections 2.3 and 4.1.
We agree that we could have used the radiances as input parameters, but it would have been technically much more complicated for us, as we would have needed to download the full AIRS and IASI spectra (2378 channels and 8461 channels, respectively) and then choose the most relevant channels for the training of the ANN models. As we have trained models for different cloud types and clear sky, for the reasons described in our manuscript (for example Cb is very rare,

with less than 10%), we would have needed anyway the CIRS data for the distinction of scene types. Furthermore it would have needed additional care to adapt the IASI spectral channels to those of AIRS in order to use the same models. As our funding is very limited (the three co-authors worked each only 6 months on this project), we decided to use the information which was easily available and which is also used in combination for further studies.